# Non-local validated parametrization of an agent-based model of local-scale *Taenia solium* transmission in North-West Peru

Francesco Pizzitutti[1]*, Gabrielle Bonnet[2], Eloy Gonzales-Gustavson[3], Sarah Gabriël[4], William K. Pan[5,6], Ian W. Pray[7], Armando E. Gonzalez[8], Hector H. Garcia[9,10], Seth E. O'Neal[7,9], for the Cysticercosis Working Group in Peru[¶]

1 Geography Institute, Universidad San Francisco De Quito, Quito, Ecuador, 2 Centre for the Mathematical Modelling of Infectious Diseases (CMMID), London School of Hygiene & Tropical Diseases, London, United Kingdom, 3 Tropical and Highlands Veterinary Research Institute, Universidad Nacional Mayor de San Marcos, Lima, Peru, 4 Department of Veterinary Public Health and Food Safety, Ghent University, Ghent, Belgium, 5 Nicholas School of Environment Durham, Duke University, Durham, NC, United States of America, 6 Duke Global Health Institute, Durham, Duke University, Durham, NC, United States of America, 7 School of Public Health, Oregon Health and Science University Portland State University, Portland, OR, United States of America, 8 School of Veterinary Medicine, Universidad Nacional Mayor de San Marcos, Lima, Peru, 9 Center for Global Health Tumbes, Universidad Peruana Cayetano Heredia, Lima Peru, 10 Cysticercosis Unit, National Institute of Neurological Sciences, Lima, Peru

¶ Membership of the Cysticercosis working group in Peru is provided in the Acknowledgments.
* francesco.pizzitutti@gmail.com

**Data Availability Statement:** The minimal dataset underlying the results described in the model are

## Abstract

The pork tapeworm, *Taenia solium*, is the cause of a preventable zoonotic disease, cysticercosis, affecting both pigs and humans. Continued endemic transmission of *T. solium* is a major contributor of epilepsy and other neurologic morbidity, and the source of important economic losses, in many rural areas of developing countries. Simulation modelling can play an important role in aiding the design and evaluation of strategies to control or even eliminate transmission of the parasite. In this paper, we present a new agent based model of local-scale *T. solium* transmission and a new, non-local, approach to the model calibration to fit model outputs to observed human taeniasis and pig cysticercosis prevalence simultaneously for several endemic villages. The model fully describes all relevant aspects of *T. solium* transmission, including the processes of pig and human infection, the spatial distribution of human and pig populations, the production of pork for human consumption, and the movement of humans and pigs in and out in several endemic villages of the northwest of Peru. Despite the high level of uncertainty associated with the empirical measurements of epidemiological data associated with *T. solium*, the non-local calibrated model parametrization reproduces the observed prevalences with an acceptable precision. It does so not only for the villages used to calibrate the model, but also for villages not included in the calibration process. This important finding demonstrates that the model, including its calibrated parametrization, can be successfully transferred within an endemic region. This will enable future studies to inform the design and optimization of *T. solium* control interventions in villages where the calibration may be prevented by the limited amount of empirical data, expanding the possible applications to a wider range of settings compared to previous models.

available from the Github repository (https://github.com/oflixs/CystiAgents).

**Funding:** This study was funded by the US National Institutes of Health National Institute of Allergy and Infectious Diseases, grant number NIH R01AI141554. The funders had no role in study design, data collection and analysis, decision to publish, or preparation of the manuscript.

**Competing interests:** The authors have declared that no competing interests exist.

# Introduction

The pork tapeworm, *Taenia solium*, is a cestode parasite that infects humans with the adult stage intestinal tapeworm and pigs with metacestode cysts in body tissue. Transmission between these hosts is cyclical. Humans acquire intestinal tapeworm infection (human taeniasis, HT) by consuming undercooked pork containing *T. solium* cysts, while pigs become infected with cysts (porcine cysticercosis, PC) by consuming human feces contaminated with *T. solium* eggs produced by intestinal tapeworms. In poor rural areas, substandard sanitation, outdoor human defecation, and the practice of allowing pigs to roam free sustain transmission. Humans can also be infected with cysts (human cysticercosis) through accidental ingestion of *T. solium* eggs, a condition responsible for an estimated third of seizure disorders in endemic areas [1]. Recently, the WHO underlined the urgent need for *T. solium* control strategies to be implemented in endemic countries to address this substantial disease burden [2, 3], and for the development of transmission models to guide design of these strategies.

Many approaches to *T. solium* modeling have been published including deterministic, Reed-Frost stochastic, decision-tree, conceptual frameworks, and agent-based models (ABM) [4]. However, none has yet achieved the accuracy, credibility, and generalizability necessary to produce evidence-based recommendations for intervention strategies. Our group recently published CystiAgent, an ABM representing local-scale *T. solium* transmission in Peru that provides several important advances [5, 6]. These include a spatially explicit framework to capture local transmission heterogeneity and to allow modelling of spatially targeted interventions, an open population structure to allow parasite influx into the modeled area, and measurement of model output accuracy against real-world longitudinal data collected during control interventions. Despite these advances, there are still several challenges remaining to achieve a credible model meeting WHO objectives.

One of the main limitations with CystiAgent involves the approach that was proposed for the model calibration [5, 6]. Calibration [7] is a standard procedure used to assign values to model parameters that represent complex sequences of unknown probabilities that cannot be measured empirically. Typically, the possible parameter space is systematically sampled and corresponding model outputs are statistically evaluated to identify parameter values that provide best fit of model outputs to observed data [7]. The CystiAgent calibration approach presented in previously published studies [5, 6] relies on six parameters, which promotes overfitting and limits model credibility. This can be addressed by streamlining steps representing parasite transmission in the ABM, thereby capturing the unknown probabilities in fewer transmission parameters. Additionally, this CystiAgent calibration procedure limits a widespread model application because it is designed to be repeated whenever a new village is modeled. A pooled calibration of one single parameter set on a group of similar villages and the transferability of the resulting model parametrization to villages not belonging to the calibration group has not yet been tested, which is the approach that is being evaluated in this paper.

For *T. solium*, the primary observables that can be compared against model outputs during calibration are the prevalence of HT and of PC. Empirical surveillance data on these outcomes are scarce, however, due to the high cost of mass human stool screening and mass pig necropsy, and limited to rare settings where intensive epidemiologic research has been conducted. This lack of empirical data to calibrate the model for most endemic setting emphasizes the need of an approach to the model calibration that does not depend on availability of village-level empirical data. To design a such approach, it is useful to note that the parameters that can be used to calibrate the model can be divided in two types: non-local and local parameters. Among non-local parameters are included parameters relatively invariant among villages that share similar genetic, cultural, geographic, and climatic features, such as the probability of

human infection upon consuming pork contaminated with *T. solium* cysts or the probability of pig infection connected with the ingestion of *T. solium* eggs. Local parameter are parameters that do depend on the specific location or village like for example the proportion of people using latrines, number of pigs per household, proportion of household owning pigs etc. Limiting the calibration to only non-local parameters can improve the model transferability creating a parametrization representative of an endemic region, rather than a single village.

Our study had a twofold objective: first to update CystiAgent and then to address the challenges connected with the development of a calibrated model parametrization not dependent on a particular village. Regarding to the first objective, the descriptions of specific model modifications are presented here, including a more detailed representation of processes involved in transmission such as pig infection, human infection, human movements and intra-village distribution of home-produced pork. Among the objective of these modifications was the reduction of the number of parameters used to calibrate the model that was achieved for instance with the introduction of the actual number of cysts to represent the pig infection. While some important processes such as pig immunity and environmental factors related to transmission like the climate remain unrepresented due to a lack of data to inform these changes, the updates described here will facilitate inclusion of these processes in CystiAgent in the future as data becomes available. Concerning the development of a non-local model parametrization we present an approach in which model parameters connected with unmeasurable host-parasite interactions can be calibrated simultaneously across a set of villages. We then validated the performance of this non-local model calibrated parametrization against empirical data observed in a group of villages that did not include the villages used for calibration.

## Methods

### Purpose of the model

The main goal of the model presented here is to reproduce the dynamics of local-scale *T. solium* transmission in the typical environment of a small rural village in Northwestern Peru. Many factors are involved in the transmission of *T. solium*, including biology and behavior of humans and pigs, socioeconomic conditions of exposed populations, and factors connected with specific characteristics of the physical environment where transmission occurs. The model is designed to capture some, but not all, of the elements recognized as important in determining *T. solium* transmission dynamics. The design of the model centers around relationships connecting real-world patterns, like prevalence of human and pig infections, with characteristics and behaviors of individual households, humans, pigs and tapeworms. Certain potentially relevant processes such as pig immunity, some human behaviors related with pig meat processing and pork consumption, and the possible role of invertebrates in transmission [8], are intentionally withheld from the model and will be reconsidered at a later stage as supporting data becomes available.

### Description of modeled area and data source

All data used to inform, calibrate, and validate the model were collected in rural villages located in the province of Piura, Peru, during a trial conducted over 2014–2015. The climate in the region is arid except during a 3-month rainy season. Household pig raising is common, both as an income source and for in-house meat consumption, and many households allow pigs to roam freely to scavenge for food. Latrines are not universally distributed, and are often of poor quality and readily accessible by pigs. Many people defecate outdoors. Together, these conditions create high-risk conditions for *T. solium* transmission [9].

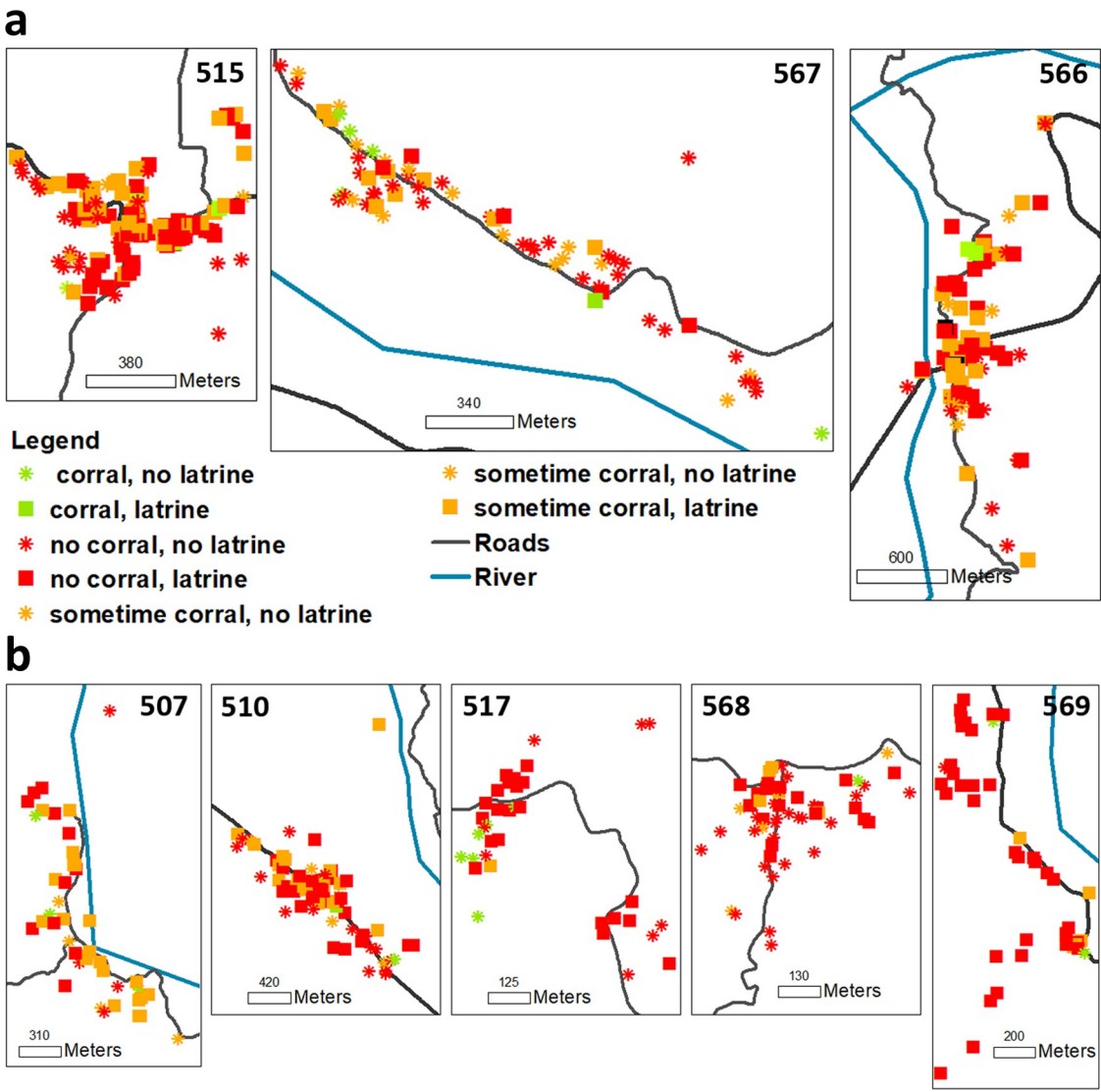

**Fig 1. Simulated villages.** Geographical distribution of households, use of corrals and latrine ownership in the villages that were used in the development of the model. Panel **a**: control villages used for model calibration. Panel **b**: intervention villages used for model validation.

The informing trial studied the effects of a community-based *T. solium* control program [10, 11], and included 5 villages that received an intervention intended to increase the community-led surveillance and pig treatment (intervention arm), and 3 villages that did not (control arm). During the trial, data were collected about household characteristics including geographical position, occupancy, infrastructure (electricity, water, latrines and adherence to its use), number of animals owned by each household, and animal-rearing practices. Individual-level data including age, sex, and education level of household members were also gathered (Fig 1; Table 1). At the end of the trial, village-level HT prevalence was determined through mass screening of human stool for presence of tapeworm eggs or antigens. PC prevalence was also evaluated at this time by purchasing as many antibody-positive pigs as villagers would agree to sell, then dissecting these in entirety to count *T. solium* cysts in each pig. Because the 3 control villages received no intervention, the final observations of HT and PC were considered

**Table 1. Simulated villages characteristics.**

| Village | Residents | Households | Pigs | Households with latrines (%) | Households raising pigs (%) | Households always using corrals for pigs (%) | Mean residents per household | Mean pigs per household (*) |
|---|---|---|---|---|---|---|---|---|
| Control villages (Used for model development, calibration, and cross-validation) | | | | | | | | |
| 515 | 240 | 68 | 90 | 20 (29.4) | 32 (47.0) | 1 (1.5) | 3.5 | 2.6 |
| 566 | 273 | 82 | 101 | 59 (71.9) | 28 (34.1) | 2 (2.4) | 3.3 | 3.3 |
| 567 | 596 | 151 | 285 | 112 (74.1) | 72 (47.7) | 6 (4.0) | 4.0 | 3.5 |
| Intervention villages (Used for validation of final calibration set-up) | | | | | | | | |
| 507 | 130 | 43 | 140 | 35 (81.3) | 19 (44.1) | 1 (2.3) | 3.0 | 7.4 |
| 510 | 280 | 79 | 142 | 55 (69.6) | 29 (36.7) | 4 (5.0) | 3.5 | 4.9 |
| 517 | 131 | 36 | 40 | 28 (77.8) | 10 (27.7) | 1 (2.8) | 3.6 | 4.0 |
| 568 | 211 | 58 | 91 | 26 (44.8) | 20 (34.4) | 0 (0) | 3.6 | 4.5 |
| 569 | 160 | 44 | 71 | 42 (95.5) | 22 (0.5) | 0 (0) | 3.6 | 3.2 |

Characteristic of villages from the 2014–2015 field trial in Piura region, Peru that were used in the development of the model

* Excludes households that do not own pigs.

to reflect natural-state of transmission and were used to calibrate the model. Data from the 5 intervention villages of the trial were used to validate the model transfer process. As showed in the original study [10], the trial interventions produced no significative effects on observed HT and PC prevalence therefore the resulting empirical data can be considered as measurements of the baseline unperturbed villages state exactly as the data from the 3 control villages. This makes the 5 intervention villages data the ideal candidate to validate the transfer process.

All the simulated village households were initialized in the model using data from their real-world counterpart, including geographical position, number and infection state of human members, presence of latrine, number of pigs, animal rearing practices like ownership and use of pig corrals, and age, sex and infection state of each pig.

## Ethic statement

The experimental protocol used to collect the data used to calibrate and validate the model presented in this paper was reviewed and approved by the Institutional Review Board at Oregon Health & Sciences University, including a waiver allowing the use of previously collected protected health information without obtaining informed consent for the purposes of this new research.

## Short description of the model

In this section will be presented an overall and concise description of the model. A detailed model description will be illustrated in the following sections. The model presented here is a development of a previously published model [5, 6] and it designed to reproduce local-scale *T. solium* transmission dynamics in the simulated villages by replicating relevant conditions in the study area during the period of data collection. Three classes of agents are included: humans, pigs and households. Several elements important for *T. solium* transmission are represented, including geographical distribution of households, human and pig population characteristics, and human and pig behavior.

Each human agent is associated with a circular defecation site area around its household. If the human agent is a tapeworm carrier, the defecation site is labeled as contaminated with T. solium proglottids and eggs. The level of contamination of the defecation site depends on the presence of a latrine in the tapeworm carrier household and on adherence to its use. When the

tapeworm carrier recovers from infection, its defecation site is considered as to be contaminated exclusively by *T. solium* eggs (no proglottids remain) until all the eggs deteriorate and are no longer viable. Pig agents may become infected when defecation sites contaminated with *T. solium* proglottids or eggs are contained in their respective roaming areas. When PC infection occurs, the model computes the number of *T. solium* cysts the pig develops. Each pig agent is assigned a home range area centered on the household owning the pig. The exposure of pig agents to contaminated sites is separated in the model between in-home-range and out-home-range exposures, with in-home-range exposure being predominant [12].

Both human and pig populations in the model are dynamic. The human population approximates natural deaths, births, emigration, and immigration rates observed in the region of Piura during the simulation period. Human agents can emigrate and definitively leave the village or leave for short period of time and then came back to the village. New human agents, that can be affected by taeniasis, are periodically introduced permanently or temporarily into the simulation to mimic people immigrating in the village or visiting from neighboring areas, respectively. The pig population approximates herd size and slaughter age, observed in the trial as well as pig imports and exports observed in the Piura region. Each household selects a female pig as the breeding sow and lets it reproduce and generate new pig agents. Additionally, each household regulates the size of its herd, trying to maintain the number of pigs as close as possible to the target herd size, defined as the number of pigs owned by the corresponding real-world village household at the time of the trial. Regulation may be done through pig export, sale to another village household, or at-home slaughter. When a pig agent is exported, it is removed from the simulation. When a pig agent is slaughtered at home, several pork portions, depending on its weight, are produced. If the pig is infected by porcine cysticercosis, its *T. solium* cysts are distributed to the resulting pork portions. After that, the pork portions are distributed first to members of the household that slaughtered the pig, and subsequently to members of other village households. All pork portions are consumed in the model. Pork portion are distributed freely to village human agents without considering any stratification based on wealth connecting the ability to pay with pork consumption. Each cyst that a human agent ingests is associated with a probability of developing a tapeworm. Human infection with multiple tapeworms is not allowed. Human cysticercosis [13] and related seizure disorders are not included in this model.

## Process overview and scheduling

The time step of the model is one week. Spatial and time scales represented in the model are consistent with the model purposes and with spatial and time scales of human and pig movements in the real world, as well as the time scales of *T. solium* development. Any process involved in *T. solium* transmission that occurs outside these scales is not represented explicitly. For example, the minute details of *T. solium* development and maturation within its hosts are not represented explicitly, nor are the effects of long-term, large-scale, processes like sanitation improvements or climate change.

## Model detailed description

The processes represented in the model are separated into four interdependent modules, including household, human, pig and demographic modules. The following sections provide a detailed description of these modules, except for the demographic module that is described in the S1 File. Flow charts representing processes in the household, human, and pig modules are also available in the S2 File.

**Table 2. Household module parameters.**

| Parameter name | Value | Description and reference |
|---|---|---|
| *slaughterAgeMean (slaughterAgeSd)* | 2.27 (0.515) months | Lognormal Mean (lognormal standard deviation) of the distribution of pig slaughtering ages [5, 6]. |
| *pigSold* | 0.514 | Probability of selling a pig that is removed from the herd [5, 6]. |
| *pigsExported* | 0.731 | Probability of exporting a pig (as opposed to selling it within the village), if it is sold [5, 6]. |
| *pigImportRateHousehold* | 0.00248 pigs per household per week | Average number of pigs imported per household per week [5, 6]. |
| *weightAgeConversion* | 0.36 kg per weeks | Conversion factor from pig age to live weight [15]. |
| *ppC* | 0.358 kg per person per week | Average weekly pork consumption Peru in rural areas (Agriculture and Irrigation Ministry of Peru) |
| $c_{entrails}$, $c_{muscle}$ | 0.155, 0.562 | Conversion factor from the live weight of a pig to entrails and muscle skin and bones [15] |

## Household module

The household module governs processes pertaining to pig slaughter, sale, and distribution of pork portions to human agents. The names of model parameters presented for the first time in this paragraph and in the following paragraphs, are shown in italic. Their values together with the reference source, are shown in Table 2.

**Herd size regulation.** When the simulation starts, the number of pig agents owned by each village household is equal to the household's target herd size, defined as the number of pigs owned by their real-world household counterpart. One pig in each herd is identified as a breeding sow; as pigs are born from breeding-sow farrows, the herd size increases. A pig is removed from the herd if 1) the herd size exceeds the target herd size, and 2) the pig age has reached or exceeded its minimum slaughtering age. The minimum slaughtering age of a pig agent is set at the beginning of its life from a log-normal distribution of parameters with mean "*slaughterAgeMean*" and standard deviation "*slaughterAgeSd*" for non-breeding pigs. Breeding sows live 4 times longer than other pig agents, which reflects practices in rural Northwestern Peru.

**Decisions regarding sale or slaughter of a pig.** When a pig agent is to be removed from the herd, the household first decides whether it will be sold. The decision to sell the pig is based on the probability expressed by the parameter "*pigSold*". If the pig is sold, it may be exported out of the village with the probability "*pigsExported*"; exported pigs are removed from the simulation altogether. The remaining sold pigs are sold to other households in the village that, in turn, can decide to export them with the probability "*pigsExported*" or to slaughter them at home immediately after purchase. Households can also decide to import a pig from outside the simulation with a weekly import frequency specified by the parameter "*pigImportRateHousehold*". Imported pig agents are assigned the same probability of having cysticercosis as the average probability of PC in the simulated villages at baseline, representing import exclusively from neighboring areas with similar levels of *T. solium* transmission.

**Definition and allocation of pork portions.** When a pig is slaughtered at home it is converted into pork portions (Fig 2). The live weight of the pig is calculated based on the age of the pig using the proportionality factor "*weightAgeConversion*". Live weight is then converted into pork portions, defined as a share of the pig carcass that corresponds to the weight of pork consumed in Peru per person per week (model parameter "*ppC*"). Pork portions may be obtained from entrails (liver, spleen, lungs and intestines) that never contain *T. solium* cysts [14], or from muscle, bones and skin that are usually sold and consumed together, that can contain *T. solium* cysts in the muscle part [15]. The number of portions is calculated using the

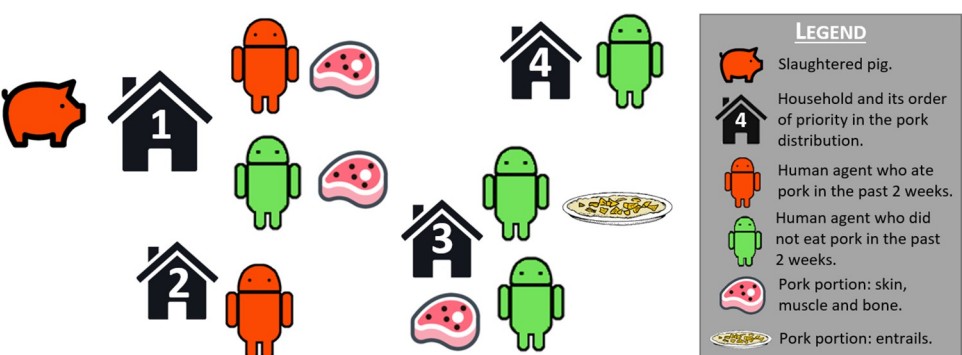

**Fig 2. Pork distribution.** The pig that was slaughtered belongs to household 1. Pork portions are distributed to all household members, including those that recently ate pork. Other households are then prioritized according to their distance to household 1. Humans in these households that ate pork in the past week do not receive pork. If there are no portions left, humans from the furthest households do not receive pork either (in this case, household 4). Skin, muscle and bone portions may contain cysts while entrails never do.

equation:

$$nP_p = \frac{\text{weight} \cdot c_p}{ppC} \text{ for } p = \text{entrails, muscles} \tag{1}$$

ppC is the weekly average consumption of pork per person. $c_{\text{entrails}}$ and $c_{\text{muscle}}$ are conversion factors from the total live pig weight to pork, while $nP_{\text{entrails}}$ and $nP_{\text{muscles}}$ are the number of portions from entrails and from muscle, bones and skin respectively [15]. If the slaughtered pig is infected with *T. solium*, its cysts are distributed uniformly to portions that are made of muscle, bones, and skin, but not to portions from entrails. Pork portions are then distributed for consumption, with a maximum of one portion allocated to each human agent in the village. Distribution always starts with members of the household that slaughtered the pig. Pork portions are then distributed to members of other households in the village starting with the household's closest neighbors. If a human agent that does not belong to the household that slaughtered the pig already received a pork portion in the same or previous week, that agent is not included in the distribution.

It is known that different part of the carcass generally present different cyst densities [16]. However, the model assumption of uniform distribution of cysts in the muscle, bone and skin portions is justified by the fact that no data is available about the way different parts of the carcass are distributed in pork portions during at-home pig slaughterings. Additionally, there is no information about different rates of allocation of distinct parts of the carcass to people with dissimilar individual characteristics in the village. These lacks of data make not possible, in the model, to effectively represent the connection between variable risk of individual exposures and different densities of cysts in distinct pork portions obtained from the same slaughtered pig.

**Human module.** The human module governs processes pertaining to human behaviors. All the parameter values and relative references are shown in Table 3.

Human agents are created in each household at the beginning of the simulation, in the same number as in the corresponding real-world village households. When a human agent eats a pork portion, that agent may develop a tapeworm if the pork portion is contaminated by *T. solium* cysts. The model expresses the probability of human infection upon consumption of a contaminated pork portion as the sum of the probabilities of being infected by each cyst contained in that portion. The probability of human infection connected with the ingestion of a

**Table 3. Human module parameters.**

| Parameter name | Value | Description and reference |
|---|---|---|
| *pHumanCyst* | (*) | Probability of human infection (taeniasis) upon ingestion of a single *T. solium* cyst. Non-local calibration parameter. |
| *adherenceToLatrine* | (*) | Probability of adherence to latrine use. Local calibration parameter. |
| *contRadiusMean (contRadiusSd)* | 26.3 (1.7) m | Mean (standard deviation) household contamination radius [6, 17]. |
| *decayMean* | 0.125 | Weekly survival probability of *T. solium* eggs dispersed in the environment [18]. |
| *tnIncubation* | 8 weeks | Tapeworm incubation time [19] |
| *tnlifespanMean (tnLifespanSd)* | 104 (50) weeks | Mean (standard deviation) of the normal distribution of tapeworm lifespan [19] |
| *travelIncidence* | $2.3 \times 10^{-4}$ | HT incidence in travel destinations [10] |
| *travelProp* | 0.423 | Proportion of village households with a traveler. [Unpublished local survey] |
| *travelFreq* | Every 8 weeks | Average frequency of trips by human travelers. [Unpublished local survey] |
| *travelDuration* | 1.75 weeks | Average duration of trips by human travelers. [Unpublished local survey] |

* Calibration parameter: see the "Model calibration and validation" section.

single cyst is represented by the model parameter "*pHumanCyst*". Being the human infection upon consuming infected pork a process that was calibrated in this and also in the past version of CystiAgent [5, 6], the introduction of cysts to represent pig infection, allowed us to reduce the number of calibration parameters. The reason is that before the model was using two distinct parameters to represent human infection: the probability to be infected upon consumption of lightly and heavily infected pork portions that was based on the corresponding description of pig infection in terms of three states: susceptible, lightly and heavily infected. In contrast, in this version of CystiAgent we use pHumanCyst as the only parameter.

Latrines are assigned to households using data from the field survey (Table 1). Without a latrine in their own household, human agents always defecate outdoors. If the household owns a latrine, the parameter "*adherenceToLatrine*" specifies the proportion of indoor defecation for household members. The position of the human agents outdoor defecation site is fixed throughout the simulation and is selected randomly inside a contamination area associated to each household when the agent is created or born or when the human agent arrives in the village if it is an immigrant or a traveler. The household contamination area is a circular area around the household whose radius is extracted from a log-normal distribution with mean "*contRadiusMean*" and standard deviation "*contRadiusSd*". Defecation sites of tapeworm carrier agents are considered contaminated with both *T. solium* proglottids and eggs after the intestinal tapeworm reaches maturity (parameter "*tnIncubation*"). The level of contamination of the defecation site depends on the presence of a latrine in the tapeworm carrier household and on adherence to its use. During the household census, study teams characterized latrines generally as either being in good state or in disrepair with respect to the ability of pigs to have potential access to human feces. Within the model, we therefore assumed that when latrines characterized as being in a good state were used, these would completely prevent pig access to human feces. If the tapeworm carrier household does not own a latrine the level of contamination is 1 otherwise it falls between 1 and 0 depending on the adherence to latrines use in the village. Pig agents may become infected upon exposure to defecation sites contaminated with *T. solium* proglottids or eggs, and when PC infection occurs, the model computes the number of *T. solium* cysts the pig develops. The model considers that proglottids do not persist in the

environment because they are immediately ingested by pig agents that actively and continuously scavenge for human feces. When the tapeworm dies, the site is no longer contaminated with proglottids, but eggs remain present until they deteriorate and are no longer infective to pigs. The longevity of *T. solium* eggs dispersed in the outdoor environment is expressed as a weekly probability of egg decay represented by the parameter "*decayMean*".

Human resident in the simulated village can die and can leave the village permanently when they emigrate. Similarly, new resident human agents can be born or immigrate into the village. In this region, people often travel between villages for short-term employment or to visit relatives [5]. Accordingly, one human agent per household in a selected group of households is designated as a traveler. Travelers leave the village at regular intervals and spend time in other endemic areas where they can be infected by *T. solium*. While the human agent is traveling, it is removed from the village and, if it is a tapeworm carrier, no new contamination of its defecation site occurs. The incidence of HT during travel, "*travelIncidence*", is obtained as the ratio between average HT prevalence observed in the simulated villages and the average duration of travels. Other model parameters describing human travel from simulated villages to external villages include "*travelProp*", the proportion of village households with a human agent traveler, "*travelFreq*", the frequency of travel of human travelers, and "*travelDuration*", the average duration of these trips. The model also considers travelers from external villages to simulated villages. Travelers from external villages are created as if they were traveling from a village with the same endemic characteristics as simulated villages, using the same values for *travelProp*, *travelFreq* and *travelDuration*. HT prevalence among travelers from external villages is set as the average HT prevalence from the three simulated villages. When an external traveler arrives in a village, it is allocated to a random household and a defecation site is created inside the household contamination area. If the traveler is a tapeworm carrier, the defecation site is considered contaminated by *T. solium* eggs and proglottids as for any other human agent in the village, for the duration of the traveler stay. Since human movement data was not available for the specific simulation villages, human movement rates were obtained from surveys conducted in neighboring areas under the assumption that the rates are similar across villages in the Piura area (see Table 3).

**Pig module.**  The pig module governs processes pertaining to pig behavior (roaming patterns), pig reproductive biology (time to maturity, gestation time, time between parity), *T. solium* cyst maturation (time to infectivity), and the probability of acquiring PC infection upon ingestion of a single egg or a proglottid. All the parameter values and relative references are shown in Table 4.

**Table 4. Pig module parameters.**

| Parameter name | Value | Notes and references |
|---|---|---|
| *homeRangeMean (homeRangeSd)* | 44.25 (1.7) m | Lognormal mean (Lognormal Sd) of the home range radius of pigs [17] |
| *pigPHomeArea* | 0.93 | Proportion of time a pig spends in its home-range area |
| *pigProglotInf* | (*) | Probability of developing a single cyst in pig upon ingestion of a single *T. solium* proglottid. Non-local calibration parameter |
| *pigEggsInf* | (*) | Probability of developing a single cyst in pig as consequence of being exposed to *T. solium* eggs from a single defecation site. Non-local calibration parameter |
| *immatureCystsPeriod* | 10 weeks | [5] |
| *sexualMaturityAge* | 26 weeks | [15] |
| *gestationTimeLength* | 16 weeks | [15] |
| *betweenParityPeriod* | 12 weeks | Personal communication Armando E. Gonzalez |

* Calibration parameter: see "Model calibration and validation" section

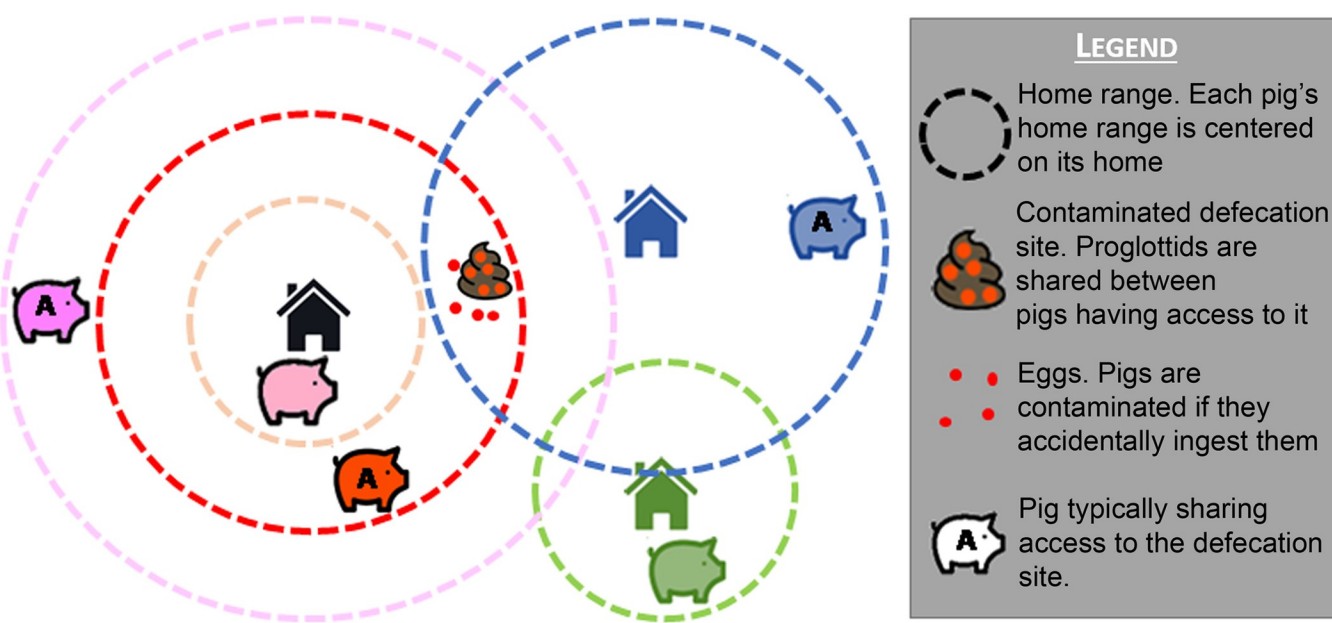

**Fig 3. Estimation of contamination levels (pig module).** The proglottids in the defecation site represented on the graph are typically shared between the pigs whose home range includes the site. The out-home range contamination term accounts for the occasional wandering of other pigs near the site. Disseminated eggs can be seen at the defecation site.

Each pig agent is assigned a home range area centered on the household to which the pig belongs. The radius of the home range is determined at the beginning of the life of each pig agent, extracting it from a log-normal distribution of parameters "*homeRangeMean*" (lognormal mean) and "*homeRangeSd*" (lognormal standard deviation). The pig agent is exposed to *T. solium* proglottid and egg contamination both inside and outside of its home range (Fig 3). The model parameter "*pigPHomeArea*" specifies the proportion of time a pig spends in its home range.

**Weekly exposure to contaminated sites.**   The total weekly contamination (C) level to which a pig is exposed is expressed as the sum of in-home-range contamination $C_{in}$ and out-home-range contamination $C_{out}$ as follows:

$$C = C_{in} \cdot pigPHomeArea + C_{out} \cdot (1 - pigPHomeArea) \qquad (2)$$

The model evaluates the weekly contamination levels inside and outside (subscripts *in* and *out*, respectively) a pig home-range by summing the contributions from proglottids and from *T. solium* eggs (subscripts *E* and *P*, respectively) in each area:

$$\begin{cases} C_{in} = C_{Pin} + C_{Ein} \\ C_{out} = C_{Pout} + C_{Eout} \end{cases} \qquad (3)$$

It is assumed that proglottids remain in feces rather than being dispersed in the environment. Given the competition for human feces that takes place between pigs, with all feces being rapidly eaten, an individual pig's exposure depends both on the number of sites infected with proglottids it has access to, and on the number of pigs that have access to these same sites. Hence, the contamination level associated with each defecation site contaminated with *T. solium* proglottids is divided by the number of pigs that have access to it ($nPig_i$ being the number of pigs having access to the i-th defecation site). Contamination also depends on latrine

availability and use by individuals with taeniasis. Hence, in the calculation of the in and out home-range contaminations from proglottids, $C_{Pin}$ and $C_{Pout}$, the number of defecation sites of tapeworm carriers not having latrines ($P_{NLin}$ and $P_{NLout}$) inside and outside the home-range respectively is added to the number of defecation sites, inside and outside the home-range, of tapeworm carriers having latrines in their households ($P_{Lin}$ and $P_{Lout}$), multiplied by the factor (1 −*adherenceToLatrine*) to account for intermittent use of these latrines. This value is multiplied by 7 (days/week) to represent the number of defecations per human agent per model time step:

$$\begin{cases} C_{Pin} = 7 \cdot \left[ \sum_{i=1}^{P_{Nlin}} \frac{1}{nPig_i} + (1 - adherenceToLatrine) \cdot \sum_{j=1}^{P_{Lin}} \frac{1}{nPig_j} \right] \\ C_{Pout} = 7 \cdot \left[ \sum_{i=1}^{P_{Nlout}} \frac{1}{nPig_i} + (1 - adherenceToLatrine) \cdot \sum_{j=1}^{P_{Lout}} \frac{1}{nPig_j} \right] \end{cases} \tag{4}$$

Egg contamination also depends on latrine availability and use. However, it is distinct from contamination with proglottids as eggs are assumed to be dispersed in the environment and not systematically targeted for consumption. The contamination factor for eggs therefore reflects incidental exposure to eggs rather than systematic consumption. The scaling factor $(N_h-1) \cdot D$ is introduced to account for the extension of the area over which eggs from contaminated defecation sites are dispersed with the density factor D. D is the ratio between the area of the union of village household contamination areas and the sum of these areas (see Eq (6)) and $N_h$ is the number of households. Latrine use also affects eggs contamination. The computation of in and out home-range contaminations from *T. solium* eggs, $C_{Ein}$ and $C_{Eout}$, is done by summing the number of defecation sites contaminated with eggs from humans not using latrines ($E_{NLin}$ and $E_{NLout}$) inside and outside the home-range respectively, with the number of defecation sites contaminated with eggs, inside and outside the home-range, of humans $E_{Lin}$ and $E_{Lout}$), multiplied by the factor (1—*adherenceToLatrine*).

$$\begin{cases} C_{Ein} = E_{NLin} + E_{Lin} \cdot (1 - adherenceToLatrine) \\ C_{Eout} = \frac{E_{NLout} + E_{Lout} \cdot (1 - adherenceToLatrine)}{(N_h - 1) \cdot D} \end{cases} \tag{5}$$

$$D = \frac{\text{Area}(U_H)}{\text{Area}(S_H)} \tag{6}$$

**New cysts developed by a pig in a week.** Contamination levels are subsequently used to compute the number of cysts that will infect a pig each week. The model represents pig infection as a stochastic process in which the number of cysts infecting the pig is determined by the contamination levels to which the pig is exposed. Specifically, the numbers of cysts derived from exposure to contamination levels from proglottids and eggs $C_P$ and $C_E$ are generated as realizations of two Poisson distribution of parameters "*pigProglotInf*" and "*pigEggsInf*", multiplied for the respective level of contaminations.

Eqs (4) and (5) express the exposure risk inside and outside a pig's home-range connected with levels of contamination, while Eq (2) reflects the proportion of time a pig spends inside vs. outside of its home range. Hence, the number of cysts nC infecting the pig in a week is

expressed as:

$$nC = (C_{PH} \cdot p_{PH} + C_{EH} \cdot p_{EH}) \cdot \text{pigPHomeArea} + (C_{PO} \cdot p_{PO} + C_{EO} \cdot p_{EO}) \cdot (1 - \text{pigPHomeArea}) \tag{7}$$

Here $p_{PH}$ and $p_{PO}$ are two distinct realizations from a Poisson distribution of parameter "*pigProglotInf*", while $p_{EH}$ and $p_{EO}$ are extracted from a Poisson distribution of parameter "*pigEggsInf*". After the pig agent is infected, the new *T. solium* cysts pass through an immature stage before being infective to humans. The length of the immature stage for cysts is specified by the model parameter "*immatureCystsPeriod*".

**Pig reproduction.** A single female pig agent can be designated as a breeding sow by the household to which it belongs, and only breeding sows can become pregnant and generate new pig agents; the number of new pig agents is random and follows a uniform distribution between 1 and 3. Age of sexual maturity for a breeding sow is specified by the parameter "*sexualMaturityAge*" and the length of the gestation period by the parameter "*gestationTimeLength*". After giving birth, the sow can be impregnated after a time indicated by the parameter "*betweenParityPeriod*".

## Model calibration

**Overview of calibration approach and summary statistics used.** We used the likelihood-free method of Approximate Bayesian Computation (ABC) to calibrate the model, tuning unknown parameters to fit model output to data observed in the three rural Peruvian villages from the control arm of the trial. We followed a Sequential Monte Carlo (SMC) approach to ABC [20] to improve the low acceptance rate of a simple ABC rejection sampler algorithm [21]. The SMC consists in defining a sequence $\{\varepsilon_i\}$ of strictly decreasing ABC tolerances, running a simple ABC rejection sampler for each tolerance $\varepsilon_i$, then exploring the parameter space through importance sampling indicated by the posterior distribution obtained at the previous stage. The first rejection sampler starts with a uniform prior distribution of calibration parameters, defined over a large and reasonable range of parameter variation. The posterior distribution estimated at step i is then used through importance sampling as prior distribution for step i+1, and the procedure is repeated until convergence. To calibrate our model, each ABC-SMC stage sampled 120,000 input parameter points with tolerances defined to accept 0.00015% of sampled points at each step. This tolerance was chosen as a trade-off to balance the speed of ABC-SMC convergence and the accuracy of calibration vector space sampling. The points in the calibration parameter space generating model outputs showing the lowest distance from empirical data were accepted, and provided a sample of the posterior calibration parameter distribution.

We introduced two different summary statistics, $\mathbf{S_{noNecro}}$ and $\mathbf{S_{Necro}}$ for each simulation village defining two distinct calibration setups (noNecro and necro calibration setup), to allow comparison of calibration including and excluding empirical data on the number of cysts per pig.

$$\mathbf{S_{noNecro}}\,(\mathbf{x}) = (HT(\mathbf{x}),\ PC(\mathbf{x}))$$
$$\mathbf{S_{necro}}\,(\mathbf{x}) = (HT(\mathbf{x}),\ PC(\mathbf{x}),\ \mathbf{necro}(\mathbf{x})) \tag{8}$$

Where x is a system state, HT and PC are the prevalence for human taeniasis and porcine cysticercosis in the study villages, respectively, and **necro** is the proportion of infected pigs

with different cyst burdens found during necropsy in the village pig population i.e.:

$$\mathbf{necro}_i = \frac{N_i}{N_{pig}} \tag{9}$$

$N_{pig}$ is the total pig population and $N_i$ the number of pigs carrying nC cysts with nC $\epsilon$ ($10^i$, $10^{i+1}$] for i = 0, . . .,5.

The distances between observed ($x_0$) and simulated (x) summary statistics in a single village are defined as follows:

$$\begin{aligned} \rho_{noNecro}(\mathbf{x},\ \mathbf{x}_0) &= \rho_{prev}(\mathbf{x},\ \mathbf{x}_0) \\ \rho_{necro}(\mathbf{x},\ \mathbf{x}_0) &= \rho_{prev}(\mathbf{x},\ \mathbf{x}_0) + \rho_{necro}(\mathbf{x}, \mathbf{x}_0) \end{aligned} \tag{10}$$

where

$$\begin{aligned} \rho_{prev}(\mathbf{x},\ \mathbf{x}_0) &= \frac{(HT(\mathbf{x}) - HT(\mathbf{x_o}))^2}{HT} + \frac{(PC(\mathbf{x}) - PC(\mathbf{x_o}))^2}{PC(\mathbf{x_o})^2} \\ \rho_{necro}(\mathbf{x},\ \mathbf{x}_0) &= \sum_{i=0}^{5} \frac{(necro_i(\mathbf{x}) - necro_i(\mathbf{x_o}))^2}{necro_i(\mathbf{x_o})^2} \end{aligned} \tag{11}$$

Here $\rho_{prev}$ accounts for the differences between observed and simulated HT and PC prevalence rates and the term $\rho_{necro}$ relates to the difference between the observed and simulated shares of pigs per number of cysts. When the denominator in Eq (11) is zero, as in the case of $necro_i$ for several values of i in all the simulated villages, then the scaling factor is set to be equal to 1. The observed summary statistic data for the three study villages are shown in the S3 File.

**Calibration distinguishing local from non-local parameters.**    Four parameters connected with unclear or insufficiently documented aspects of *T. solium* transmission, "*pHumanCyst*", "*pigProglotInf*", "*pigEggsInf*", and "*adherenceToLatrine*", were tuned during the calibration process, in order to obtain a model whose output approximates observed epidemiological conditions based on defined biologic, environmental and socioeconomic parameters. These calibration parameters were either non-local (values are invariant across villages in a single region) or local (values vary across villages in a single region). Given the genetic homogeneity of human and pig populations in the three simulation villages, we assumed that parameters describing how *T. solium* interacts with its hosts to be non-local. The parameters "*pHumanCyst*", "*pigProglotInf*", and "*pigEggsInf*", which describe the probability of human infection upon consumption of infected pork, and the number of cysts a pig develops upon exposure to *T. solium* proglottids and eggs, respectively, are considered non-local parameters. The calibration parameter "*adherenceToLatrine*", describing human agent adherence to the use of latrines, is the only local calibration parameter because the adherence to latrine use was never assessed through filed survey and it can, in principle, strongly depend on the village household number and geographical density. Each calibration parameter was provided a reasonable starting variation range (see S3 File); the marginal prior distribution for the first stage of ABC-SMC is a uniform distribution over that range.

To calibrate the model with local and non-local calibration parameters, a single global ABM was created including the entire group of the three simulation villages. In this global ABM, there are no interactions or human or pig movements between these three simulated villages, although pig import and export, human travel, and migration to/from places outside the three villages was retained. This global ABM was calibrated following the ABC-SMC approach, allowing the local calibration parameter to take different values for different villages. The i-th

sampling point in the input parameter space is a vector taking the same values for non-local parameters and three distinct values for the local parameter:

$$p_{i,j} = (g_{i,1}, g_{i,2}, g_{i,3}, l_{i,j}) \; j = 1, 2, 3 \tag{12}$$

Where $g_{i,1}$, $g_{i,2}$ and $g_{i,3}$ are the i-th realizations of the three non-local calibration parameters and $l_{i,j}$ is the i-th realizations of the local calibration parameter for the j-th village. The global ABC calibration distance is defined as the sum of the three village distances defined in Eq (10), weighted by the relative share of the total simulated human and pig populations residing in the village:

$$\rho_{\text{global}} = \sum_{i=1}^{3} \frac{P_i}{P_{\text{global}}} \cdot \rho_i \tag{13}$$

$\rho_i$ is the ABC distance for the i-th village while $P_i$ and $P_{\text{global}}$ are the populations of humans plus pigs in the i-th village and in all three villages, respectively. The ABC-SMC calibration was considered as having converged when the addition of a new stage did not produce a decrease in the minimum ABC distance.

**Cross-validation of calibrated model.**  We assessed the fit of calibration parameters using cross-validation [22]. The cross-validation method uses the ABC simulations pool of the final calibration setup stage to assess the accuracy of posterior distributions estimation. The final stage ABC simulation pool is composed by the entire set of simulations run for that stage of the calibration process. To build the cross-validation of a calibration setup, 100 calibration parameters points are selected at random from the final stage setup simulation pool and the corresponding model simulation outputs are designed as pseudo empirical data. The ABC is then applied to the remaining simulations of the pool to estimate the parameter values that produced the best approximations of each of the 100 selected pseudo observed dataset. Since the true parameter values corresponding to the pseudo empirical data are known, cross-validation between estimated and real parameter values are used to assess the accuracy of the posterior distribution estimation, by the ABC setup, of each specific calibration parameter.

**Simplified calibration of only non-local parameters.**  Model transferability would be improved if all parameters were non-local. We therefore evaluated a third calibration setup, in addition to the necro and noNecro setups, whereby the parameter "*adherenceToLatrineUse*" was removed from the set of calibration parameters. In this calibration, "*adherenceToLatrineUse*" was set to 0.55 for all villages based on the average calibrated value obtained for noNecro setup for the three calibration villages. The resulting simplified calibration setup relies only on three non-local calibration parameters: *pHumanCyst*, *pigEggsInf* and *pigProglotInf*.

**Validation of the simplified calibration method.**  Based on the calibration results presented below, we selected the simplified calibration setup to assess its performance when the model is transferred to a new group of villages and validated against the resulting new set of empirical data. We transfer the simplified calibration setup to the five villages of the intervention arm of the field trial [10, 11] (Table 1 and Fig 1), and compared the relative errors (REs) of model output for HT and PC prevalence against the REs obtained during simulation of the three villages of the control arm. REs between and observed prevalence $\text{Prev}_{\text{obs}}$ and a simulated prevalence $\text{Prev}_{\text{sim}}$ are defined as: $|\text{Prev}_{\text{obs}} - \text{Prev}_{\text{sim}}| / \text{Prev}_{\text{obs}}$.

## Model software and simulations outcomes calculations

The model is implemented in a software architecture that includes an ABM core developed using MASON [23], a free, Java-based, discrete-event multi-agent platform, and our in-house

Java software code to run the ABC calibration. The cross-validation ABC analysis was done using the abc R package [22, 24]. Each simulation run in this study started with 3,500 burn-in weekly time-steps followed by 10,000 production time-steps. Only production steps were used to sample, every 100 weeks, the summary statistics. For the ABC-MCS calibration process only one simulation per calibration point was run. For comparison with observed data, simulated summary statistics were averaged over the outcomes of a pool composed by 128 repetitions of the same model simulation. Simulations were run in parallel on the Duke Compute Cluster (DCC) at Duke University, USA. Every simulation run, for three villages, required approximately, in average, 70 seconds of one Intel Core i7-800 processor.

The program code used for running the simulations and model calibrations is available on a GitHub repository at https://github.com/oflixs/CystiAgents. A minimal dataset underlying the results described in this paper is available under the directory "inputFile" of the program code repository. Nevertheless, given the need to protect the confidentiality and privacy of research subjects, the individual-level data used as inputs to the model are only available, in the public repository, as de-identified and aggregated.

## Results

### Calibration setups and posterior distributions

Posterior distributions for the model's calibration parameters resulting from ABC-SMC calibration are calculated initially considering two distinct ABC setups. The first setup (necro setup) uses the ABC distance $\rho_{necro}$ of Eq (10) that corresponds to the summary statistic $\mathbf{S_{necro}}$ of Eq (8). The second ABC setup (noNecro setup) uses the distance $\rho_{noNecro}$ of Eq (10) hence not including the observed proportion of infected pigs with different cyst burdens to the calibration summary statistics.

The calibration process produced a remarkable narrowing of the posterior marginal distributions, respect to the prior marginal distribution shown in S3 File, Table 2, for the non-local calibration parameters of the noNecro calibration setup, and to a lesser extent of the necro setup (Fig 4). For local parameters, the marginal posterior distribution is narrowed for village 567 and somewhat narrowed for village 566, noNecro but for village 515 the narrowing is limited. Table 5 shows the values of HT and PC prevalence compared with observed prevalences. REs of simulated prevalences compared to the observed values show that the noNecro calibration setup produced a better approximation of empirical prevalences as compared to the necro setup, except for PC prevalence in village 515 and HT prevalence in village 566 for which the relative errors are equals.

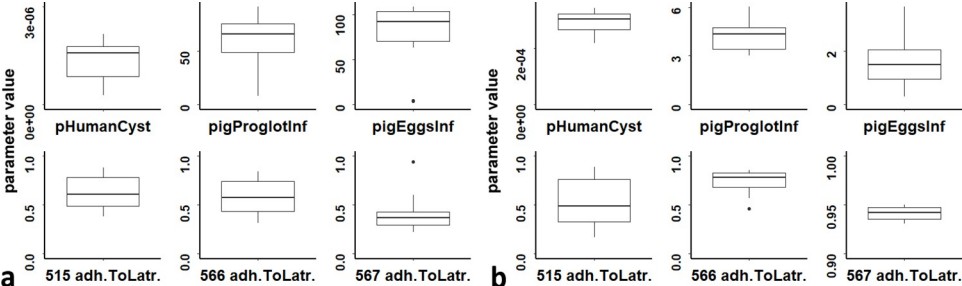

**Fig 4. ABC posteriors distribution.** Posterior distributions of model calibration parameters for the two calibration setups: **panel a:** necro calibration setup (fourth ABC-SMC stage). **Panel b:** noNecro calibration setup (third ABC-SMC stage). Horizontal bold lines indicate the posterior distribution median, boxes cover the parameter values above the first quartile and below the third quartile and vertical lines show maximum and minimum values. Black dots show outliers.

**Table 5. Calibration results.**

| Village—calibration setup | HT (RE) | PC (RE) |
|---|---|---|
| 515 –observed | 0.024 | 0.33 |
| 515—necro | 0.0013 (95%) | 0.25 (24%) |
| 515—noNecro | 0.024 (0%) | 0.24 (27%) |
| 566—observed | 0.0097 | 0.15 |
| 566—necro | 0.0014 (86%) | 0.32 (113%) |
| 566—noNecro | 0.018 (86%) | 0.2 (33%) |
| 567 –observed | 0.02 | 0.16 |
| 567—necro | 0.0014 (93%) | 0.31 (94%) |
| 567 –noNecro | 0.016 (20%) | 0.16 (0%) |

Values of simulated HT and PC prevalence of noNecro ABC setups compared to the observed prevalence. The simulated values are the product of simulation run using the third calibration stage necro, and the fourth stage noNecro setups. The Relative Errors (RE) is defined as: $|\text{Prev}_{obs}\text{-Prev}_{sim}|/\text{Prev}_{obs}$.

Concerning the reproduction of the observed shares of pigs having different numbers of cysts, in principle those empirical data should be fitted better by the necro setup that uses a summary statistic in which the shares of pigs per number of cysts are explicitly included (Eqs (10) and (11)). This is reflected by the lower values obtained when the necro ABC distance $\rho_{necro}$, defined in Eq (10), is calculated using the best run produced by the necro calibration setup as compared to the best run of the noNecro setup (Table 6). Nevertheless, a visual comparison (Fig 5), shows that the specific characteristics of empirical curves of shares of pigs per number of cysts are not captured in either the noNecro or necro calibration setup. For example, the empirical curve for village 515 shows a pronounced peak for a number of cysts burden around $10^4$, while for village 566 the curve is zero for a number of cysts greater than 100. Those features, together with the observed value for number of cysts equal to 1000 for village 567, are not reproduced either by the necro or noNecro calibration setups.

## Cross-validation

Cross-validation, using the same tolerance as the third stage of noNecro ABC-SMC, was successful in reproducing the true value of the three non-local parameters for the selected pseudo empirical data (Fig 6). However, the ability of cross-validation to reproduce the local calibration parameter was generally poor. The performance was lowest for village 515, suggesting that, in this village, the variable "*adherenceToLatrine*" has limited influence on the outcome of the model. This is not unsurprising, as relatively few households in this village have latrines, so contamination is primarily driven by individuals to whom the variable "*adherenceToLatrine*" does not apply. The fit is better, though still imperfect, for village 566, with village 567

**Table 6. ABC distances.**

| ABC setup | ABC dist. | Tot dist. ($\rho_{prev}+\rho_{necro}$) | Prevalence dist. ($\rho_{prev}$) | Necro dist. ($\rho_{necro}$) |
|---|---|---|---|---|
| necro | 1.29 ($\rho_{necro}$) | 1.29 | 1.06 | 0.74 |
| noNecro | 0.19 ($\rho_{noNecro}$) | 1.42 | 0.19 | 1.23 |

Minimum ABC distances for necro and noNecro calibration setups. The column 'ABC dist.' shows the actual ABC distance used for calibration. The column 'Tot dist' shows the sum of $\rho_{prov}$ and $\rho_{necro}$ as defined in Eq (11). The $\rho_{necro}$ distance for noNecro calibration setup is showed here only to be compared with the necro calibration setup but is not included in the ABC distance of noNecro calibration setup.

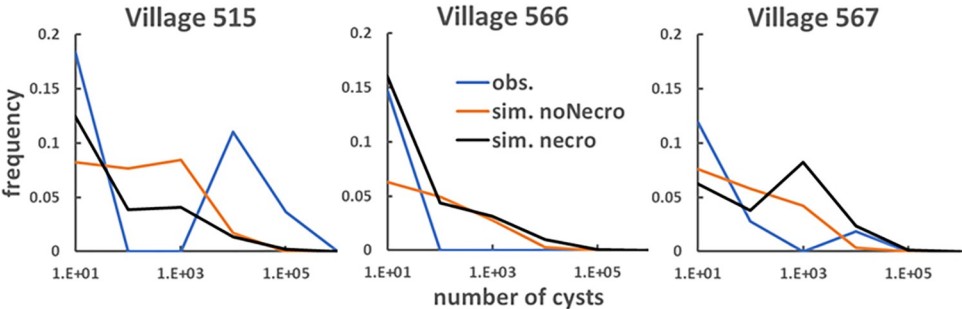

**Fig 5. Calibration results.** Proportion of infected pigs with different cyst burdens estimated using the necro and noNecro calibration setups as compared with observed data for control group villages 515, 566 and 567. The simulated curves are the product of the average over 750 runs using the minimum ABC distance parameter sets for both the necro and noNecro setups.

performing best. These two villages have relatively high shares of the population having latrines (between 70 and 80%). The cross-validation graphs suggest that, in these villages, the variable "*adherenceToLatrine*" has a greater importance than in village 515, but that several parameters set with different values for "*adherenceToLatrine*" can still be a good fit for experimental data. This suggests that the considered summary statistics did not fully capture the model parameter describing adherence to latrine use. Results of the cross-validation for the necro setup are shown in the S3 File.

## Simplification of the calibration

The simplified calibration setup relies only on three non-local calibration parameters: *pHumanCyst*, *pigEggsInf* and *pigProglotInf*. This setup does not calibrate local parameters and do not include the data from necroscopies. The third stage of SMC-ABC simplified calibration for this setup, shows a marginal posterior distribution narrowed respect to the prior distribution (see S3 Fig 2 in S3 File) and can reproduce (Table 7) empirical data better than the necro setup

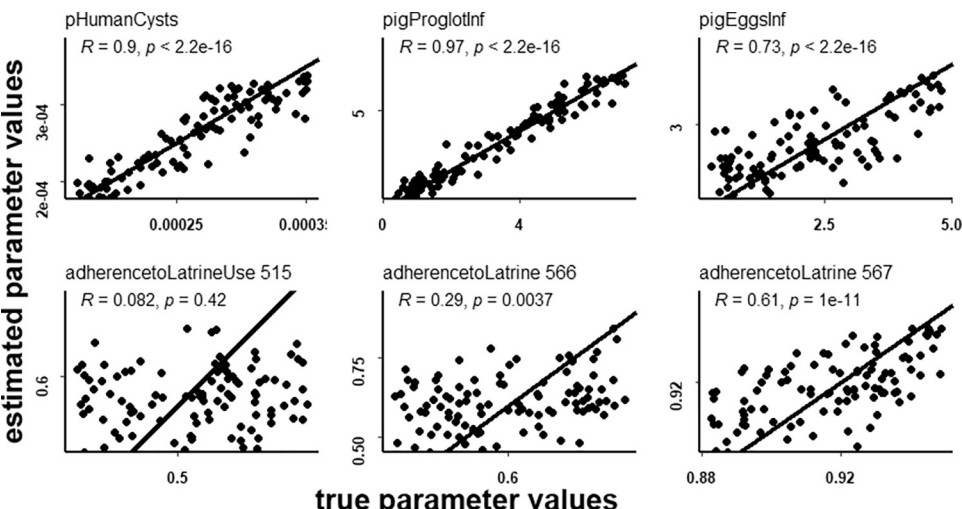

**Fig 6. Cross-validation noNecro setup.** Cross-validation for 100 randomly selected simulations from the noNecro calibration setup, third stage. Estimated parameter values are compared to true values for the 6 calibration parameters with the identity line plotted as reference. The correlation coefficient between estimated and true parameter values is shown for each parameter together with the corresponding p value.

**Table 7. Simplified setup calibration results.**

| Village—calibration setup | HT (RE) | PC (RE) |
|---|---|---|
| 515—simplified | 0.013 (45%) | 0.15 (54%) |
| 566—simplified | 0.013 (34%) | 0.14 (7%) |
| 567—simplified | 0.012 (40%) | 0.28 (75%) |

Values of simulated HT and PC prevalence for simplified ABC setup with RE. The simulated values are the product of simulations run using the minimum ABC distance parameter sets for the third calibration stage. The Relative Errors (RE) is defined as: $|Prev_{obs}\text{-}Prev_{sim}|/Prev_{obs}$. The REs are calculated using observed data from Table 5.

except for PC prevalence in village 515, and better than the noNecro calibration setup in village 566. The average REs over the three village for the simplified calibration setup are 40% and 45% for HT and PC, respectively. Those averages are not too far from the corresponding values for the noNecro setup (35% and 20% for HT and PC, respectively) and are better than for the necro setup (91% and 77%). As expected, the calibrated model parameter values for the noNecro and simplified calibration setups (Table 8) are different with more pronounced differences in the values of parameters pHumanCyst and pigProglotInf. The former parameter is associated with noNecro and simplified posterior distributions that do not overlap as it can be noted comparing Fig 4 and S3 Fig 2 in S1 File. Cross-validation plots shown in Fig 7 show that the simplified model calibration can estimate precisely the calibration parameters.

## Validation of the simplified calibration

Simulations of the five villages of the intervention arm to which the model was transferred, using the simplified calibrated model parametrization, produced average REs equal to 48% and 46% for HT and PC respectively (Table 9). These results are very close to the average REs of control villages (40% and 45% for HT and PC, respectively, see Table 7). Only for village 507 the particularly low observed values of PC prevalence produced a remarkably high RE value. The result of the simulations demonstrates that the parametrization obtained from the global calibration procedure presented here can be effectively exported to similar villages without any additional adjustment.

## Discussion

The objective of this study was to update CystiAgent, an ABM of *T. solium* local-scale transmission, and to explore the extent to which its credibility and transferability can be improved by reducing the number of parameters requiring calibration, and by eliminating the need to perform calibration for each modeled village. We evaluated an approach in which parameters representing unmeasurable host-parasite interactions could be calibrated simultaneously across a group of villages in an endemic region, and explored the performance of these non-

**Table 8. Parameter from the best fit runs.**

| Setup | non local parameters | | | | local parameter (*) |
|---|---|---|---|---|---|
| | pHumanCyst | pigEggsInf | pigProglotInf | Village | adherenceToLatrine |
| noNecro | $3.41\ 10^{-4}$ | 0.94 | 3.86 | 515, 566, 567 | 0.35, 0.85, 0.95 |
| simplified | $2.08\ 10^{-4}$ | 1.17 | 3.29 | 515, 566, 567 | |

Parameter from the best fitting run of noNecro and simplified calibration setups. Non-local parameters take the same values for all the villages and the local parameter is included only for the noNecro calibration setup.

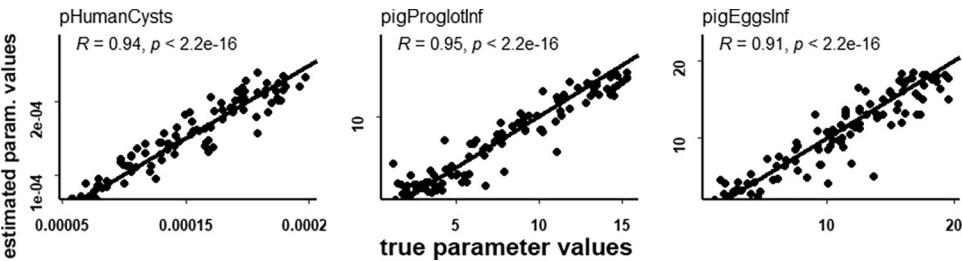

**Fig 7. Cross-validation, simplified setup.** Cross-validation for 100 randomly selected simulations from the simplified calibration setup, third stage. Estimated parameter values are compared to true values for the 3 calibration parameters with the identity line plotted as a reference. The correlation coefficient between estimated and true parameter values is shown for each parameter together with the corresponding p value.

local parameters through validation against empiric disease prevalence data from a different set of villages in the same endemic region of northwestern Peru. We found that the updated ABM, calibrated with only 3 non-local parameters, was able to adequately simulate baseline disease transmission in the source dataset for villages used during calibration, as well as in a distinct dataset from other villages that were not part of the calibration process. This finding suggests that the model is likely to replicated baseline levels of transmission with adequate accuracy for other villages within the same region presenting characteristics of exposed human populations, their socio-economics status, pig breeds and environmental and climatologic conditions similar to those of the villages used to calibrate the model. This development represents an important step towards the creation of an accurate, credible, and transferable model which can be used to inform intervention strategies to control or eliminate *T. solium*.

We updated several important aspects of *T. solium* transmission in this second version of CystiAgent ABM. The first update concerned the improvement of the process by which pig agents acquire cysticercosis. The prior version, as well as other published ABM of *T. solium* transmission [25], relied upon a categorization of pigs into groups of either light, heavily, or uninfected pigs, as well as simplistic one-step change of pig infection state after an exposure to environmental *T. solium* contamination. In this new version of CystiAgent, pig infection and pork contamination states are based upon accumulation of individual cysts. The number of

**Table 9. Calibration validation results.**

| Village | HT (RE) | PC (RE) |
|---|---|---|
| 507—observed | 0.011 | 0.046 |
| 507—simplified | 0.012 (9%) | 0.1 (117%) |
| 510—observed | 0.019 | 0.11 |
| 510—simplified | 0.012 (37%) | 0.15 (36%) |
| 517—observed | 0.0 | 0.15 |
| 517—simplified | 0.012 (na%) | 0.2 (33%) |
| 568—observed | 0.053 | 0.19 |
| 568—simplified | 0.013 (75%) | 0.23 (21%) |
| 569—observed | 0.045 | 0.19 |
| 569—simplified | 0.012 (73%) | 0.14 (26%) |

Values of simulated HT and PC prevalence rates calculated for the five intervention villages compared to the observed prevalence. The simulated values are the product of the average over 750 runs using the parametrization of the simplified calibration model. The Relative Error (RE) is defined as: $|\text{Prev}_{obs}-\text{Prev}_{sim}|/\text{Prev}_{obs}$.

cysts infecting the pig determines the grade of infection, and these cysts are transferred from slaughtered pigs to pork portions, creating a direct link between exposure of pigs to environmental contamination levels and the number of cysts to which human individuals are exposed upon pork consumption. The introduction of cysts makes the ABM more flexible, as it allows direct representation of aspects of *T. solium* transmission that are not yet included in the model. For example, pig immunity can be introduced in a straightforward way, connecting mechanisms for immunity against oncospheres, immature cysts, and mature cysts, through inhibition of cyst development and cyst degeneration. Additionally, it allows the inclusion of data describing the cyst burden distribution in the village pig population to calibrate the model, as in the necro calibration setup we presented here.

Another feature that differentiates this version of CystiAgent from its predecessor, is the greater detail in the representation of human movement, and the introduction of natural human demographic dynamics. Human movements on different scales of time and space can strongly influence the endemic state of a village [26] and are specially important once control measures are implemented. CystiAgent now supports not only short-term human movements of villagers to external destinations, but also arrival of short-term residents from endemic areas. Long term emigration and immigration from endemic and non-endemic areas are also now represented in the model, as are natural demographic dynamics of birth and death. Parameters can be readily adjusted in the demographics module, facilitating model transferability to new endemic areas with different population dynamics.

Other ABM updates include an improved representation of pig infection outside the home range area, a greater detail in pork distribution and sale that was needed to reproduce more closely the corresponding real world processes, and introduction of pig population demographics that will be useful in introducing pig immunity in future updates.

The approach to calibration was also modified in the new version of CystiAgent. Instead of calibrating the model separately for each village, a non-local strategy to calibrate the model simultaneously for more than one village, was used. For the villages in northwestern Peru considered in this version, and for many other endemic areas, the characteristics of exposed human populations, their socio-economics status, pig breeds, and environmental and climatologic conditions, can be considered as homogeneous. This suggests that, if all the location specific characteristics of each village in a model, like household geographical positioning, distribution of corral ownership and use, latrine ownership and use, are represented correctly and by means of known parameters, the remaining parameters connected with transmission host-parasite interactions (the non-local calibration parameters of this paper), if unknown, can be estimated through a non-local calibration that does not depend on any specific village. The simulation results presented here showed that, if designed appropriately, a *T. solium* local-scale transmission model, with the same set of calibrated parameter values, can reproduce the baseline epidemiological conditions for most villages in a homogeneous endemic area demonstrating the concept of transferability of non-local calibrated parameters through simulations of the five "new" intervention villages.

Another difference between the prior and current versions of CystiAgent concerns the source of calibration data. The calibration of ABMs of *T. solium* transmission requires an estimate for the observed HT and PC prevalence. As noted, those data are scarce, location-specific, highly variable in time and difficult to obtain. Moreover, it is almost impossible to directly measure HT and PC prevalence without disrupting the baseline endemic state. The approach taken in the prior version of CystiAgent was to estimate baseline PC and HT prevalences based on observed serum antibody patterns in the pig population observed in villages not included in the study and calibrate the models tuning parameters to fit these estimates [27]. However, the accuracy of the transferability of these estimations within the simulated villages

is unknown, thereby complicating the interpretation of the ABM performance, as it is calibrated based on this data. In this updated version of CystiAgent version, the ABM is calibrated using empiric HT and PC data measured [10] through mass human treatment and subsequent stool copro-antigen assessment joined with light microscopy to detect eggs in the stool (HT) and mass necropsy of the pig population (PC) of simulated villages. The resulting validation process is based on gold standard methods and direct observations that provides a more reliable estimation of model parameters.

The outcomes of model calibration presented here suggest that the SMC-ABC used to tune the model to empirical data was successful in parametrizing the model. The narrowing of the posterior distributions demonstrates that the calibration process is effectively estimating the posterior distributions for the unknown model parameters. Moreover, the successful cross-validation demonstrated that the posterior distributions are estimated accurately. The only non-local calibration parameter, *adherenceToLatrineUse*, was successfully eliminated in the simplified calibration setup, allowing for an effective calibration with only three non-local calibration parameters.

The goodness of fit for the estimation of empirical data is important to establish model credibility. The relative errors (REs) on the estimation of observed HT and PC prevalence averaged over the three villages were 35% and 20% for the noNecro setup, and 40% and 45% for the simplified calibration setup. Although those errors are not insignificant, their interpretation should be approached with caution. These REs are based upon a single measurement of data observed in a system in which dynamic fluctuations of transmission levels occur over time even in an undisturbed state producing empirical measurement of prevalence at a single point in time associated with high uncertainty. Therefore, the ABM should be expected to approximate empiric values rather than to reproduce them exactly. Furthermore, if the error is instead calculated based on the number of infected humans and pigs, the approximation by which the calibrated model is reproducing the empirical data appears better, particularly in the case of HT prevalence. For example, for the smallest village in the control group (village 515), the infection of a single additional human with taeniasis produces a change in RE of about 18% for the simplified calibration setup with respect to the observed baseline rate. Therefore, the RE on the approximation of HT prevalence for village 515 corresponds to only 2.6 tapeworm carriers on a total population of 240 residents. Similarly, the model approximates the observed HT prevalence rates with REs that correspond to a difference of just 0.9 and 4.7 tapeworm carriers in populations of 273 and 596 residents for villages 566 and 567 respectively. In the case of empirical PC prevalence, the simplified calibration setup REs corresponds to a difference of 16.3, 3.0 and 34 infected pigs for villages 515, 566 and 567, respectively for pig populations of 90, 101 and 285.

The unsatisfactory reproduction of the observed proportion of infected pigs with different cyst counts can be attributed to various factors. First, given that the overall number of necropsies in each village was small (15, 16 and 42 necropsies for villages 515, 566 and 567, respectively), the high uncertainty on the empirical values makes it similarly challenging to reproduce these data during the calibration process. Since mass necropsies in the undisturbed state are exceedingly rare and expensive to conduct, improvement of data quality for this is unlikely. On the other hand, inclusion of pig immunity in a subsequent ABM version is feasible, and may improve the ability of the model to accurately reproduce the distribution of cysts among pigs in a modeled village. However, there is currently insufficient data available to achieve this goal. Experiments that detail the probability of pig infection upon ingestion of *T. solium* eggs, controlling for factors such as dose, pig age, prior exposure and infection, and presence of maternal antibodies, are needed.

The lack of a full representation of pig immunity is not the only limitation of this new version of CystiAgent. The model still relies on a representation of environmental *T. solium* contamination based on the count of defecation sites contaminated with proglottids and eggs and not on the actual densities of eggs to which the pigs are exposed. Due to the absence of data to effectively estimate the density of eggs produced by tapeworm carriers when defecating outdoor, it is not yet possible to improve model representation of environmental contamination. Another process that is considered important in determining *T. solium* transmission and that was not represented in the model because of the limited amount of available data is the influence of seasonal dynamics on transmission. Furthermore, even if data are not available for the Piura district, for other areas in Peru an increase in pork consumption was observed at the end of the month of July, during the Peruvian national holidays and, to a limited extent, during Christmas holydays [15]. These changes, connected with cultural and religious practices, may have a strong impact on intervention effectiveness [28] and will be included in the model when more data will be available.

We anticipate conducting studies, in the Piura district, in the near future, on pig immunity, environmental contamination with eggs, human and pig, movements patterns relevant to *T. solium* transmission, infection prevalence of pigs imported into the villages pattern of pork distribution to the village members. While we also anticipate exploring the performance of the CystiAgent against empirical data gathered through interventions in Zambia, a mature development strategy to transfer the model to other endemic areas presenting conditions sufficiently different from those observed in northwestern Peru is still needed. This challenge will need to be met in order to use CystiAgent as a universal tool to support the design and testing of disease control strategies as envisioned by the WHO.

## Conclusions

The new ABM and the new, non-local, approach to model calibration introduced here extend the previously developed models of *T. solium* transmission by creating a flexible simulation framework that can reproduce the baseline epidemiological conditions of endemic villages in rural areas of northwestern Peru. The model is flexible enough to be extended to additional nearby endemic areas in the region, representing a step toward the development of a universal tool to conduct *in silico* experiments informing the design and optimization of *T. solium* control interventions.

## Supporting information

**S1 File. Demographic module.**
(DOCX)

**S2 File. ABM flow diagrams.**
(DOCX)

**S3 File. Additional data and figures.**
(DOCX)

## Acknowledgments

Membership of the Cysticercosis working group in Peru: Robert H. Gilman, MD, DTMH (Johns Hopkins University Bloomberg School of Public Health, Baltimore, MD); Manuela Verastegui, PhD; Mirko Zimic, PhD; Javier Bustos, MD, MPH (Universidad Peruana Cayetano Heredia, Lima, Peru); and Victor C. W. Tsang, PhD (Coordination Board): Silvia Rodriguez,

MSc; Isidro Gonzalez, MD; Herbert Saavedra, MD; Sofia Sanchez, MD, MSc, Manuel Martinez, MD (Instituto Nacional de Ciencias Neurologicas, Lima, Peru); Saul Santivanez, MD, PhD; Holger Mayta, PhD; Yesenia Castillo, MSc; Monica Pajuelo, PhD; Luz Toribio; Miguel Angel Orrego, MSc (Universidad Peruana Cayetano Heredia, Lima, Peru); Maria T. Lopez, DVM, PhD; Cesar M. Gavidia, DVM, PhD; Ana Vargas-Calla, DVM (School of Veterinary Medicine, Universidad Nacional Mayor de San Marcos, Lima, Peru); Luz M. Moyano, MD; Ricardo Gamboa, MSc; Claudio Muro; Percy Vilchez, MSc (Cysticercosis Elimination Program, Tumbes, Perú); Sukwan Handali, MD; John Noh (Centers for Disease Control, Atlanta, GA); Theodore E. Nash, MD (NIAID, NIH, Bethesda, MD); Jon Friedland (Imperial College, London, United Kingdom).

Leading author: Hector H. Garcia email: hgarcia@jhsph.edu

## Author Contributions

**Conceptualization:** Francesco Pizzitutti, Gabrielle Bonnet, Eloy Gonzales-Gustavson, Sarah Gabriël, William K. Pan, Ian W. Pray, Armando E. Gonzalez, Hector H. Garcia, Seth E. O'Neal.

**Data curation:** Francesco Pizzitutti, Eloy Gonzales-Gustavson, Seth E. O'Neal.

**Formal analysis:** Francesco Pizzitutti, Ian W. Pray, Seth E. O'Neal.

**Funding acquisition:** Seth E. O'Neal.

**Investigation:** Francesco Pizzitutti, Seth E. O'Neal.

**Methodology:** Francesco Pizzitutti, Gabrielle Bonnet, Sarah Gabriël, Seth E. O'Neal.

**Project administration:** Seth E. O'Neal.

**Software:** Francesco Pizzitutti, Ian W. Pray.

**Supervision:** Francesco Pizzitutti, Sarah Gabriël, William K. Pan, Seth E. O'Neal.

**Validation:** Francesco Pizzitutti, Seth E. O'Neal.

**Writing – original draft:** Francesco Pizzitutti, Seth E. O'Neal.

**Writing – review & editing:** Francesco Pizzitutti, Gabrielle Bonnet, Eloy Gonzales-Gustavson, Sarah Gabriël, William K. Pan, Ian W. Pray, Seth E. O'Neal.

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
