## [Decision Letter · Decision Letter 0]

12 Jul 2022

PONE-D-21-27051Non-local validated parametrization of an agent-based model of local-scale Taenia solium transmission   in North-West PeruPLOS ONE

Dear Dr. Pizzitutti,

Thank you for submitting your manuscript to PLOS ONE. After careful consideration, we feel that it has merit but does not fully meet PLOS ONE’s publication criteria as it currently stands. Therefore, we invite you to submit a revised version of the manuscript that addresses the points raised during the review process.

Could the authors address the points raised by the reviewers, giving special attention for the comments regarding the methodology used in the study and the clarity of the text.  

We look forward to receiving your revised manuscript.

Kind regards,

Marcello Otake Sato, Ph.D., D.V.M.

Academic Editor

PLOS ONE

Journal Requirements:

2. In ethics statement in the manuscript and in the online submission form, please provide additional information about the patient records/samples used to for validation and calibration of your model. Specifically, please ensure that you have discussed whether all data/samples were fully anonymized before you accessed them and/or whether the IRB or ethics committee waived the requirement for informed consent. If patients provided informed written consent to have data/samples from their medical records used in research, please include this information.

4. Thank you for stating the following financial disclosure: "This study was funded by the US National Institutes of Health National Institute of Allergy and Infectious Diseases, grant number NIH R01AI141554."

Please state what role the funders took in the study.  If the funders had no role, please state: "The funders had no role in study design, data collection and analysis, decision to publish, or preparation of the manuscript.

6. Please amend your list of authors on the manuscript to ensure that each author is linked to an affiliation. Authors’ affiliations should reflect the institution where the work was done (if authors moved subsequently, you can also list the new affiliation stating “current affiliation:….” as necessary).’

7. One of the noted authors is a group or consortium 'Cysticercosis Working Group'. In addition to naming the author group, please list the individual authors and affiliations within this group in the acknowledgments section of your manuscript. Please also indicate clearly a lead author for this group along with a contact email address.

8. Please ensure that you refer to Figures 1, 2, 3, 6 and 7 in your text as, if accepted, production will need this reference to link the reader to the figure.

9. Please upload a new copy of Figure 3 as the detail is not clear. Please follow the link for more information: https://blogs.plos.org/plos/2019/06/looking-good-tips-for-creating-your-plos-figures-graphics/" https://blogs.plos.org/plos/2019/06/looking-good-tips-for-creating-your-plos-figures-graphics/

Additional Editor Comments (if provided):

First of all, my apologies for the time taken for reviewing the MS. It was a really difficult task to find suitable reviewers.

Reviewers' comments:

Reviewer's Responses to Questions

**Comments to the Author**

1. Is the manuscript technically sound, and do the data support the conclusions?

Reviewer #1: Yes

Reviewer #2: Yes

2. Has the statistical analysis been performed appropriately and rigorously? 

Reviewer #1: I Don't Know

Reviewer #2: Yes

3. Have the authors made all data underlying the findings in their manuscript fully available?

Reviewer #1: Yes

Reviewer #2: Yes

4. Is the manuscript presented in an intelligible fashion and written in standard English?

Reviewer #1: Yes

Reviewer #2: Yes

5. Review Comments to the Author

Reviewer #1: This paper represents potentially an important step in developing a transferrable T. solium transmission model, however I have several major comments/clarifications to raise, as the manuscript is hard to follow (particularly the results) in the present form, and some important questions regarding the methodology. See attached PDF for full comments.

Reviewer #2: In this paper the authors presented an new agent based model of local-scale T. solium and a new, non-local, approach to the model calibration to fit model outputs to observed human taeniasis and pig cysticercosis prevalence. The findings of the authors showed that the modeling in-silico approach may be applied in other endemic areas which is quite interesting. I believe that their modeling design is sound and logical. I would like to ask the authors to please check Lines 148, 244, 329, 546 (ERROR! Reference source not found)

6. PLOS authors have the option to publish the peer review history of their article (what does this mean?). If published, this will include your full peer review and any attached files.

Reviewer #1: No

Reviewer #2: No

---

## [Author Response · Author response to Decision Letter 0]

26 Jul 2022

Authors answers to reviewers comments

Overall comments: 

This paper represents potentially an important step in developing a transferrable T. solium transmission model, however I have several major comments/clarifications to raise, as the manuscript is hard to follow (particularly the results) in the present form, and some important questions regarding the methodology. 

Intro 

Comment 1: Can the authors expand on the following sentence, as its is not clear how this would be achieved (reducing the number of transmission parameters), and whether this is relevant to this paper (I don’t think this was looked at in this paper)? 

“This can be addressed by streamlining steps representing parasite transmission in the ABM, thereby capturing the unknown probabilities in fewer transmission parameters.” (lines 89-91) 

For example, in the first objective (outlined in 109-11), is the purpose also to streamline representation of parameters, and if so, this should be made clear in the objectives. 

Authors answer: We completely agree. We expanded the first objective description to make clear that a streamline representation of parameters is among the study objectives and we added an explanation of that in the methods, section “Human module”.

Comment 2: The following sentence is not very clear, please rephrase/restructure: 

“This makes village-level calibration impractical converting it in a substantial barrier to the transferability of CystiAgent to most endemic settings.” (lines 99 -101) 

Authors answer: Yes, this phrase was not very clear. We changed the phrase with: “This lack of empirical data to calibrate the model for most endemic setting emphasizes the need of an approach to the model calibration that does not depend on availability of village-level empirical data. Among… “

Comment 3: please can the author break up the sentence 114 – 119, as this sentence makes overall sense but is a little hard to follow. 

Authors answer: OK sentence divided in two parts

Methods 

Comment 1. What about seasonal differences in transmission (if influenced by the dry or 3-month rainy season indicated by the authors on lines 137-138). For example, Braae et al. 2014 found higher probability of free-ranging in the dry season compared to wet season and therefore different exposure risk (also possibly perturbing stable endemic transmission dynamics which most models currently assume). This study reflects the situation in Tanzania, a different endemic system, but may be relevant for this system. 

Ref: Braae, U.C., Magnussen, P., Lekule, F. et al. Temporal fluctuations in the sero-prevalence of Taenia solium cysticercosis in pigs in Mbeya Region, Tanzania. Parasites Vectors 7, 574 (2014). https://doi.org/10.1186/s13071-014-0574-7

Authors answer: We agree. Seasonal differences are indeed important and, when possible, must be included in the model. Unfortunately, for the area under study, we have no data about seasonal changes of pig roaming areas (see ref 14). Other, possibly relevant changes in pork meat consumption can be connected for example with holydays and other social gatherings but also in this case no data are available. We deigned further studies to collect data on those topics that will be conducted in northern Peruvian rural areas.

Comment 2. The authors mention in lines 139 – 140 that “latrines…are often of poor quality and readily accessible by humans”. This seems like an important feature of this particular endemic system, so was information on latrine quality collected (alongside information on presence of latrine & adherence; line 158 & mapped on Fig 1) and incorporated into the model? Or do the authors assume that all latrines are of equally poor quality and therefore uniformly provide an exposure risk to free-roaming pigs? 

If the latter (this seems the case, given lines 290-291 “The level of contamination of the defecation site depends on the presence of a latrine in the tapeworm carrier household and on adherence to its use”), this should be explicitly stated in the assumptions. 2 

Authors answer: No information about latrines quality was collected during the field trial used to inform the model. We changed the text after lines 290 -291 to explain this better.

Comment 3. Regarding pig exposure (lines 183 – 188): Is the extent of pig exposure within a contaminated defecation site dependent on the duration of time spent in site, or assumed as soon as the pig agent enters a contaminated defecation site the pig will become exposed? 

Authors answer: The process is described in detail in the “Pig module – weekly exposure to contaminated sites”. We changed the sentence starting from line 183 to explain better this point: “Pig agents may become infected when defecation sites contaminated with T. solium proglottids or eggs are contained in their respective roaming areas.”

Comment 4. The authors state new human agents are “periodically introduced” into the simulation (lines 192 – 193), with the rich dataset acquired through this study, would it not be feasible (and more realistic) to accurately simulate immigration introductions (for example are there differential rates during dry vs rainy seasons)? 

Authors answer: Unfortunately, no data are available about human movements in the villages under study during the study period. For that reason, only rates from obtained from neighboring areas and from the entire Piura region were used un the study. No data about seasonal differences in human movements were available. 

Comment 5. “cysts being distributed randomly to the pork portions” (lines 201-202) is a strong assumption to include, there is good knowledge now on the carcass distribution of cysts (see Chembensofu et al. 2017), so could the authors indicate whether they have considered modelling this (assuming pork portions can be from different parts of the pig carcass?). 

The authors further state on lines 253 – 254 that “If the slaughtered pig is infected with T. solium, its cysts are distributed randomly to portions that are made of muscle, bones, and skin, but not to portions from entrails”; can the authors also explain here why cysts are not distributed to entrails (or a fuller description of what constitutes entrails would eb useful). 

Ref: Chembensofu, M., Mwape, K.E., Van Damme, I. et al. Re-visiting the detection of porcine cysticercosis based on full carcass dissections of naturally Taenia solium infected pigs. Parasites Vectors 10, 572 (2017). https://doi.org/10.1186/s13071-017-2520-y

Authors answer: We totally agree: different part of the carcass present generally different cyst densities. However, this strong assumption is justified for two reasons: first in the villages under study carcass are divided into portions at home without following any predefined meat cut patterns. The second reason justifying our assumption is that even if the right distribution of cysts in different carcass parts were followed, we have no data to connecting consumption patterns and the individual characteristics of humans in the villages. The portions of pork containing different densities of cysts would be then distributed randomly to the villagers (following the geographical criteria indicated in the manuscript) resulting in no practical effect on transmission. 

We based our assumption of no cysts in the entrails (it is now specified in the text that entrails are: liver, spleen, lungs and intestines) on papers like:

Boa ME, Kassuku AA, Willingham AL, Keyyu JD, Phiri IK, Nansen P. Distribution and density of cysticerci of Taenia solium by muscle groups and organs in naturally infected local finished pigs in Tanzania. Veterinary Parasitology. 2002;106: 155–164. doi:10.1016/S0304-4017(02)00037-7

That reported 0 cyst found in entrails. The paper cited in the comment reported 0 cysts in the entrails with the exception of the liver where only 3.7 % of cysts were found.

Comment 6. Please can the authors refer to the tables in the text where relevant and throughout, for example after describing the mean and standard deviation of the slaughter age (lines 228 – 231), referring to Table 2 here would improve clarity. 

Authors answer: Ok. We revise the entire manuscript to increase the references to tables. However please consider that each paragraph in the “Model detailed description” like for example the “Household module” that contains the lines cited in this comment starts with the sentence: “The parameters in the household module, along with their reference source,

are shown in Table 2 (or Table 3 or 4)”. The repeated reference to tables for each model parameter would make the text heavier. To make the references to tables clearer we change the opening sentence with: 

“The names of model parameters presented in this paragraph are shown in italic. Their values together with the reference source, are shown in Table 2.”. Same for other methods paragraph presenting model parameters. 

Comment 7. Given the large number of tables throughout the manuscript, I recommend condensing the tables, for example including in the same row, the overall parameters for slaughter age mean and standard deviation in Table 2 (and Table 4), rather than including two rows. The authors should try to do this across all tables to reduce the length of the methods. 

Authors answer: Thanks for the suggestion. We condensed the tables where possible.

Comment 8. After review of S1. Fig1. I am not clear where the decision process regarding sale or slaughter of pigs is included within the Household module flow chart. Indeed, it appears this is within the S2 Fig3: Pig module flow chart. Instead, which might cause some confusion when trying to match the description in the methods to these supplemental figures. 

On a further note, S1 Fig 1 in the S2 word document should be S2 Fig 1? Can the authors check in detail throughout the supplementary to ensure there are no typos such as this please. 

Authors answer: Thanks for noting that. We corrected accordingly figures S2 Fig1 and S2 Fig3 the pigs and household flow charts.

Comment 9. How valid is the assumption that “neighbouring areas have similar levels of T. solium transmission” regarding assigning the same probability of cysticercosis for imported pig agents (lines 240-242). 

Authors answer: Unfortunately, this is only a guess because not studies on prevalence of pigs imported in the study rural villages were made. We then reasonably supposed that if imported from similar rural areas pigs had to present same levels of cysticercosis prevalence. More studies are coming.

Comment 10. Can the authors provide more justification for why a maximum of one pork portion would be distributed to each human agent (lines 255 – 256), would it not be reasonable to expect different portions based on age and possible sex? 

Authors answer: We agree but we miss completely data to do that. More studies are coming.

Comment 11. Furthermore, can the explain why pork portions are distributed more widely from the initial household (lines 256-260), are these pork portions sold to the other households, or given freely? If sold, is there a probability associated with the ability to pay for the recipient household, or is this effectively captured in the pigimportRateHousehold parameter (Table 2)? 

Probabilities in Table 2??? – should they be represented by a prob distribution w/ parameters? 

Authors answer: We made the assumption, based on direct observations in the field that distribution of pork start always from the members of the household. This is another really interesting topic connecting wealth with exposure to TS infection risk. But unfortunately, again, we have not data to connect ability to pay with actual consume of pork. More studies are coming. The pork portion are given freely in the model without stratifying by household wealth. We explained this point better in the main text.

Comment 12. Should there not be a decay rate associated with proglottids (or eggs remaining within these proglottids) in the environment (295-298; “When the tapeworm dies, the site is no longer contaminated with proglottids, but eggs remain present until they deteriorate and are no longer infective to pigs”. It appears at the moment as though proglottids (or eggs within) instantly disappear upon death of the adult worm? 

Authors answer: Yes. This is because the model supposes that proglottids are always readily ingested by pigs that actively and continuously scavenge for human feces. We explained this point better in the main text.

Comment 13. How was the tolerance threshold of 0.015% chosen (line 399)? 

Authors answer: This was a mistype error! Thank for the comment. The threshold (actually 0.00015) is chosen to include in the ABC-SMC calibration selection process around the 18 best performing parameter vectors (0.00015 * 120000 calibration vectors = 18). This number is a trade-off resulting from a careful selection process balancing the speed of convergence of ABC-SMC and a sufficiently accurate sampling of the calibration vector space. We specified this in the trext.

Comment 14. On line 451, the authors write “We assessed the fit of calibration parameters using cross-validation (19). The cross-calibration method”; are these methods the same, or should the second by changed from cross-calibration to cross-validation (i.e., I don’t think cross-calibration has been mentioned before in “Calibration distinguishing local from non-local parameters” section) 

Authors answer: OK text corrected. Cross-calibration changed to cross-validation everywhere in the manuscript.

Minor (methods) comments: 

Comment 1. Infrastructure instead of infrastructures (line 145) 

Authors answer: OK

Comment 2. A reference is missing on line 148, 244; “Error! Reference source not found!” 

Authors answer: same problem of pdf conversion. We make sure that during the resubmission every reference to tables will be OK.

Comment 3. The wording in the following sentence could be improved for clarity, and Significative effects should be rephrased to significant effect (line 154): 

“Data from the 5 intervention villages, in which however, as showed in the original study (10), interventions produced no significative effects on observed HT and PC prevalence, were then used to validate the final calibration process”. 

Authors answer: OK sentence rephrased to: “Data from the 5 intervention villages of the trial were used to validate the calibration process. As showed in the original study [10], the trial interventions produced no significative effects on observed HT and PC prevalence in those villages and the resulting empirical data can be considered as baseline unperturbed data exactly as the data from the 3 control villages. This makes the 5 intervention villages data the ideal candidate to validate the calibration process.”

Comment 4. Line 178, this could be reworded to improve clarity, such as “with a circular defecation site area around its household” 

Authors answer: OK. Thanks 

Comment 5. Definitely should be definitively on line 191. 

Authors answer: OK. Thanks

Comment 6. averages should be average (line 241) 

Authors answer: OK. Thanks

Comment 7. The authors explain the calibration has been limited to “those parameters that may be relatively invariant among villages”, meaning the non-local calibration parameters (in tables 3-4) and while there is a description of “distinguishing local from non-local parameters” in the statistical methods from line 419, it would be useful if earlier in the methods the authors explicitly state that there are both local (e.g., adherence to latrines in Table 3) and non-local parameters 

Authors answer: OK introduction changed to introduce local and non-local parameters from the beginnig.

Comment 8. Can the authors provide references (or an indication that the work is from local survey work, not published?) for travelProp, travelFreq and travelDuration parameters in Table 3 please? 

Authors answer: That is exactly the case: an unpublished local survey. Table changed to indicate this. 

Comment 9. The sentence on line 367 to 370 regarding development of cysts is not clear, please can the authors break this into two sentences and re-think the wording to improve 4 clarity. This will also help to improve understanding of sentence 373 to 374 regarding parameters definitions of pigProglotInf and pigEggsInf. 

Authors answer: OK. Sentence changed in “The model represents pig infection as a stochastic process in which the number of cysts infecting the pig is determined by the contamination levels to which the pig is exposed. Specifically, the numbers of cysts derived from exposure to contamination levels from proglottids and eggs CP and CE are generated as determinations of two Poisson distribution of parameters “pigProglotInf” and “pigEggsInf”, multiplied for the respective level of contaminations. “

Comment 10. Please can the authors write lognormal mean and lognormal SD under notes and references for homeRangeMean and homRangeSd in table 4. Please also change Share of time to Proportion of time for pigPHomeArea to keep consistent with the text. 

Authors answer: OK, thanks

Results (pg.27) 

Overall, some error and not easy to follow, so I would encourage the authors to think carefully about how to better present these results. There are some errors (in tables) and missing references/ lack of text to support inclusion of tables (e.g., Table 6) that make the results difficult to digest. In the results, similar to previous sections, there are a number of “(Error! Reference source not found.)” insertions, which additionally makes the results section a challenge to understand. 

Authors answer: OK we will make sure that in the new pdf that will be generated for re-submission all those missing reference will be avoided

Specific comments follow: 

Comment 1. Can the authors include y-axis labels for figure 4 please on the plots. 

Authors answer: OK 

Comment 2. I am not entirely sure what Figure 4 is showing, is this the range from many different simulations obtaining different posterior distributions, or is the box and whisker plot for each parameter showing a single posterior distribution (with the median, IQR etc for that single posterior distribution). If the latter, it would be good to also show the posterior distribution (probability density) plots in addition. 

Authors answer: Not sure to understand the comment. As clearly stated in Fig 4 caption the plots are indeed showing the actual posterior distributions for calibration parameters. We that to add a series of plot to presenting marginal posterior distributions also as a probability density distribution would not add more information increasing the troubles of text interpretation.

Comment 3. The authors also state that there is a “remarkable narrowing of the posterior marginal distributions for the non-local calibration parameters of the noNecro calibration setup” (Figure 4 panel b) compared to the prior marginal distributions (supplementary information 3); I am not clear what prior marginal distributions the authors are comparing to in the supplementary file 3 – are the authors referring to S3 Table 2; if so, please make clear in the main text? 

Authors answer: Ok. The sentence: “respect to the prior marginal distribution shown in supplementary material S3, Table 2” was added to the text 

Comment 4. There seems to be some errors regarding the Relative Error (RE) values in Table 5, for example, for the HT RE in 566, for the necro setup, this is 86% ((0.0097-0.0014)/0.0097 = 0.8556), but noNecro is also 86%, however for noNecro RE calculation, (0.0097-0.018)/0.018 = -0.4611111 (-46%), unless there is some misunderstanding here? 

Authors answer: The RE is calculated as: |Prevobs-Prevsim|/Prevobs so the denominator is the observed value like here above first case and not like here above second case where the denominator is the simulated value. We added a definition of RE in the Methods section.

Comment 5. “This is reflected by the lower values obtained when the necro distance Pnecro is calculated using the best runs produced by the necro calibration setup as compared to the best run of the noNecro setup” (lines 506 – 508). Not clear what lower values mean here, the distance between observed and simulated? 

Authors answer: Between the two setups ABC distance that is defined in equation 10. We specified this in the text.

Comment 6. There is no reference to Table 6, should this be in line 508? If this is the case, Table 6 is quite complex so the authors should explicitly indicate which elements of lines 506-508 (if these indeed refer to Table 6) refer to specific findings in Table 6, or include further text to explain this table. 

Authors answer: we are sorry for that: in our original .doc file this reference was really close to line 508. It was actually in line 509 where in the pdf it reads: error! Reference source not found. We will make sure that this reference will be correctly showed in the re-submission pdf.

Comment 7. Please can the authors include x and y-axis lables, for example, I am assuming that the y-axis is the true values and the x-axis is the estimated parameter values for simulations (and it is not clear in the figure legend lines 560 – 563)? Can the authors also explicitly include reference to Figure 6 in the Cross-validation results section (lines 544-557). 

Authors answer: Ok x and y-axis labels added. About the reference this was again a problem with the pdf rendering. In out word file the reference was in line 547. We make sure next pdf will be correct. 

Comment 8. The Cross validation necro setup scatter plots S3: Fig 1 seem to suggest that the necro setup performs far better a reproducing the local “adherenceToLatrine” parameter compared to the noNecro (especially for villages 566 and 567), is there a reason why these results were not included in the main results? 5 

Authors answer: This decision was not based on the goodness of cross-calibration results. As stated in the manuscript, the necro setup has performed worse respect to the noNecro setup in reproducing observed data. For that reason, we decide to show in the main text only the cross-validation for the setup noNecro that was then used to build the simplified setup and we decided also to move the necro setup cross-validation results, for the reader reference, in the supplementary material. 

Comment 9. Can the authors include the observed values again for each village in Table 7, or at least say in the Table legend that the RE are calculated with observed prevalence’s from table 5, and the corresponding formula. 

Authors answer: OK legend modified

Comment 10. I would suggest making really clear, in the opening of the “Simplification of the Calibration” results section (page 31.) that simplified calibration does not calibrate the local parameter (adherence to latrines) therefore does not include the NoNecro and Necro setup (so the simplified calibration results can be compared to NoNecro and Necro setups). This will save the reader from going back to the methods to make the connection that NoNecro and Necro require the adherence to latrine parameter. 

Authors answer: OK opening modified

Comment 11. While the authors highlight that the orders of magnitude are similar for the non-local parameters in the noNecro and simplified setups, which is true (Table 8), I am not convinced by the statement that the “simplified model calibration is able to estimate precisely the calibration parameters” (lines 574 – 575) – while they are not far off, particularly for the pigProglotIn probability there is quite a difference i.e. 3.86 to 11.23, so I think this statement is too strong to make. Are there uncertainty estimates available from the fitting procedure for each parameter, to then assess whether there is overlap in these uncertainty ranges (95% credible intervals) to make the comparison more valid? There is a reference missing on page 574, which might refer to Figure 7, in which case a stronger argument can be made for this statement? 

Authors answer: Again, this problem of missing tables references. The reviewer is right this missing was a reference to Figure 7. Sorry again about that. The cited sentence “simplified model calibration is able to estimate precisely the calibration parameters” was did not referred to the actual value of the parameter but to the goodness of cross-validation test. We added an explanation to the main text to make this clearer.

About an uncertainty interval a comparison of parameter pigProglotInf for noNecro in figure 4 and for simplified in figure S3 fig 2 shows that the posterior distributions did not overlap at all so the two parameter estimation are truly different. We see this not as a problem because the model is different in the two setups. We agree that 3.86 is not strictly of the same order of magnitude of 11.23 so we changed the sentence in: “As expected, the calibrated model parametrization for the noNecro and simplified calibration setups (Table 8) are different with more pronounced differences in the values of parameters pHumanCyst and pigProglotInf”

Comment 12. The authors state “The result of the simulations demonstrates that the parametrization obtained from the global calibration procedure presented here can be effectively exported to similar villages without any additional adjustment.” (lines 595 – 594) and while the average RE might similar, the variation in RE (particularly for higher REs) seems greater for the validated results to intervention villages (with max errors of 75% for HT (validated) compared to 45% for HT (calibrated) and 117% for PC (validated) compared to 75% for PC (calibrated)). I think the authors need to caveat the above statement, especially if they are interested in transferring the model to look at individual villages in a different setting. Is there for example a threshold difference in average RE that the authors would then the globally calibrated model would not be transferrable to another endemic setting? 

Authors answer: We want to observe that the max error for HT validated is 37% and not 75%. That value falls close to the RE of 45% for the calibration. We agree with the reviewer about the fact that more quantitative criteria would be useful to discriminate among good and bad calibrations. Unfortunately, this would be possible only if we were able to estimate uncertainties of empirical data. With empirical data uncertainties it would be possible to say if a calibrated model parametrization is producing or not outcome in agreement with observed data. We are drafting a paper were pig seroprevalence is used as summary statistics and empirical data uncertainties on the observed data will make a mor quantitative criteria about model transferability available. 

Discussion 

Overall, this is a good discussion analyzing the strengths and current limitations of the model. I think, as mentioned in Results comment 11., indicating whether there was overlap in credible intervals best-fitting non-local parameter (between noNecro and simplified) for example would make statements around transferability more valid, especially as the authors argue that the RE metric should be interpreted with caution (in terms of assessing how transferrable the model is, especially given the extremes in some of the REs between the calibrated vs. validated simplified model). 

Authors answer: As stated in the answer to comment 11 there is not overlap between posterior distribution of noNecro e simplified setups. However, this says nothing about transferability. It is only saying that the parameters estimated in these two simulations setup are different which is not necessarily wrong. The transfer of the model was made from the control to the intervention group of villages, same simplified simulation setup. We modified the Method “Validation of the simplified calibration method” section to explain better this point.

The authors mention that the “resulting validation process is based on gold standard methods”, which is certainly true for necropsy, although there is limited detail on the diagnostic used for taeniasis (is this just copro-antigen assessment for example, which has substantial limitations)?

Authors answer: As stated in the paper: 

Beam M, Spencer A, Fernandez L, Atto R, Muro C, Vilchez P, et al. Barriers to participation in a

760 community-Based program to control transmission of Taenia solium in Peru. Am J Trop Med Hyg.

761 2018;98(6):1748–54.

describing the trial used to inform the model, during the trial 62% of people identified by ELISA coproantigen test, had TS eggs in the stool in indicating a robust proportion of certainly truly positive HT cases.

We modified the text to say it better.

---

## [Decision Letter · Decision Letter 1]

6 Sep 2022

PONE-D-21-27051R1Non-local validated parametrization of an agent-based model of local-scale Taenia solium transmission  in North-West PeruPLOS ONE

Dear Dr. Pizzitutti,

Thank you for submitting your manuscript to PLOS ONE. After careful consideration, we feel that it has merit but does not fully meet PLOS ONE’s publication criteria as it currently stands. Therefore, we invite you to submit a revised version of the manuscript that addresses the points raised during the review process.

Dear authors, please address the comments made by the Reviewer 1.

We look forward to receiving your revised manuscript.

Kind regards,

Marcello Otake Sato, Ph.D., D.V.M.

Academic Editor

PLOS ONE

Journal Requirements:

Additional Editor Comments:

Dear authors, please address the comments made by the reviewer 1.

Reviewers' comments:

Reviewer's Responses to Questions

**Comments to the Author**

1. If the authors have adequately addressed your comments raised in a previous round of review and you feel that this manuscript is now acceptable for publication, you may indicate that here to bypass the “Comments to the Author” section, enter your conflict of interest statement in the “Confidential to Editor” section, and submit your "Accept" recommendation.

Reviewer #1: (No Response)

Reviewer #2: All comments have been addressed

2. Is the manuscript technically sound, and do the data support the conclusions?

Reviewer #1: Yes

Reviewer #2: Yes

3. Has the statistical analysis been performed appropriately and rigorously? 

Reviewer #1: N/A

Reviewer #2: Yes

4. Have the authors made all data underlying the findings in their manuscript fully available?

Reviewer #1: Yes

Reviewer #2: Yes

5. Is the manuscript presented in an intelligible fashion and written in standard English?

Reviewer #1: Yes

Reviewer #2: Yes

6. Review Comments to the Author

Reviewer #1: I provide follow-up comments below

Methods

Comment 1. What about seasonal differences in transmission (if influenced by the dry or 3-month rainy season indicated by the authors on lines 137-138). For example, Braae et al. 2014 found higher probability of free-ranging in the dry season compared to wet season and therefore different exposure risk (also possibly perturbing stable endemic transmission dynamics which most models currently assume). This study reflects the situation in Tanzania, a different endemic system, but may be relevant for this system.

Ref: Braae, U.C., Magnussen, P., Lekule, F. et al. Temporal fluctuations in the sero-prevalence of Taenia solium cysticercosis in pigs in Mbeya Region, Tanzania. Parasites Vectors 7, 574 (2014). https://doi.org/10.1186/s13071-014-0574-7

Authors answer: We agree. Seasonal differences are indeed important and, when possible, must be included in the model. Unfortunately, for the area under study, we have no data about seasonal changes of pig roaming areas (see ref 14). Other, possibly relevant changes in pork meat consumption can be connected for example with holydays and other social gatherings but also in this case no data are available. We deigned further studies to collect data on those topics that will be conducted in northern Peruvian rural areas.

Reviewer response: Thank you, it would be useful to include comment in the discussion on the potential role of seasonal dynamics, especially around changing pork consumption practices associated with holidays etc. Lightowlers & Donadeu 2017 also mention the importance of a “pork consumption calender”, so referencing this within the specific context of Northern Peru / development of the model would be really beneficial for the reader.

Comment 2. The authors mention in lines 139 – 140 that “latrines…are often of poor quality and readily accessible by humans”. This seems like an important feature of this particular endemic system, so was information on latrine quality collected (alongside information on presence of latrine & adherence; line 158 & mapped on Fig 1) and incorporated into the model? Or do the authors assume that all latrines are of equally poor quality and therefore uniformly provide an exposure risk to free-roaming pigs?

If the latter (this seems the case, given lines 290-291 “The level of contamination of the defecation site depends on the presence of a latrine in the tapeworm carrier household and on adherence to its use”), this should be explicitly stated in the assumptions. 2

Authors answer: No information about latrines quality was collected during the field trial used to inform the model. We changed the text after lines 290 -291 to explain this better.

Reviewer response: Thank you for this additional information and text in the manuscript. So, you assumed all households had latrines in the “good state” based on the household census or you assumed this was the case?

Comment 4. The authors state new human agents are “periodically introduced” into the simulation (lines 192 – 193), with the rich dataset acquired through this study, would it not be feasible (and more realistic) to accurately simulate immigration introductions (for example are there differential rates during dry vs rainy seasons)?

Authors answer: Unfortunately, no data are available about human movements in the villages under study during the study period. For that reason, only rates from obtained from neighboring areas and from the entire Piura region were used un the study. No data about seasonal differences in human movements were available.

Reviewer response: Thank you, it would be useful for the authors to explicity state that human movement data was not available for the specific villages therefore migration rates were utilised from neighbouring areas (under the assumption that the rates are similar across villages in the Piura area?.

Comment 5. “cysts being distributed randomly to the pork portions” (lines 201-202) is a strong assumption to include, there is good knowledge now on the carcass distribution of cysts (see Chembensofu et al. 2017), so could the authors indicate whether they have considered modelling this (assuming pork portions can be from different parts of the pig carcass?). The authors further state on lines 253 – 254 that “If the slaughtered pig is infected with T. solium, its cysts are distributed randomly to portions that are made of muscle, bones, and skin, but not to portions from entrails”; can the authors also explain here why cysts are not distributed to entrails (or a fuller description of what constitutes entrails would eb useful).

Ref: Chembensofu, M., Mwape, K.E., Van Damme, I. et al. Re-visiting the detection of porcine cysticercosis based on full carcass dissections of naturally Taenia solium infected pigs. Parasites Vectors 10, 572 (2017). https://doi.org/10.1186/s13071-017-2520-y

Authors answer: We totally agree: different part of the carcass present generally different cyst densities. However, this strong assumption is justified for two reasons: first in the villages under study carcass are divided into portions at home without following any predefined meat cut patterns. The second reason justifying our assumption is that even if the right distribution of cysts in different carcass parts were followed, we have no data to connecting consumption patterns and the individual characteristics of humans in the villages. The portions of pork containing different densities of cysts would be then distributed randomly to the villagers (following the geographical criteria indicated in the manuscript) resulting in no practical effect on transmission.

We based our assumption of no cysts in the entrails (it is now specified in the text that entrails are: liver, spleen, lungs and intestines) on papers like:

Boa ME, Kassuku AA, Willingham AL, Keyyu JD, Phiri IK, Nansen P. Distribution and density of cysticerci of Taenia solium by muscle groups and organs in naturally infected local finished pigs in Tanzania. Veterinary Parasitology. 2002;106: 155–164. doi:10.1016/S0304-4017(02)00037-7

That reported 0 cyst found in entrails. The paper cited in the comment reported 0 cysts in the entrails with the exception of the liver where only 3.7 % of cysts were found.

Reviewer response: Thank you, it would be useful for the authors to summarise the above justifications underpinning the assumption around lack of connecting portion allocations from different parts of the pig carcass to variable exposure risk.

Comment 8. After review of S1. Fig1. I am not clear where the decision process regarding sale or slaughter of pigs is included within the Household module flow chart. Indeed, it appears this is within the S2 Fig3: Pig module flow chart. Instead, which might cause some confusion when trying to match the description in the methods to these supplemental figures.

On a further note, S1 Fig 1 in the S2 word document should be S2 Fig 1? Can the authors check in detail throughout the supplementary to ensure there are no typos such as this please.

Authors answer: Thanks for noting that. We corrected accordingly figures S2 Fig1 and S2 Fig3 the pigs and household flow charts.

Reviewer response: Thank you for the revision; only minor suggestion would be to make sure the arrow on the line after the box “Designate breeding sow to be killed, replace sow” is before the intersection rather than after (otherwise appears that the decision tree does not continue to the pig selling boxes.

Comment 9. How valid is the assumption that “neighbouring areas have similar levels of T. solium transmission” regarding assigning the same probability of cysticercosis for imported pig agents (lines 240-242).

Authors answer: Unfortunately, this is only a guess because not studies on prevalence of pigs imported in the study rural villages were made. We then reasonably supposed that if imported from similar rural areas pigs had to present same levels of cysticercosis prevalence. More studies are coming.

Reviewer response: Thank you, I would suggest a short section in the discussion (or methods) to state where further studies will address these gaps/ assumptions (similar for the next comment on prevalence of imported pigs, number of distributed pork portions per household member) where the authors indicate more studies are coming.

Comment 11. Furthermore, can the explain why pork portions are distributed more widely from the initial household (lines 256-260), are these pork portions sold to the other households, or given freely? If sold, is there a probability associated with the ability to pay for the recipient household, or is this effectively captured in the pigimportRateHousehold parameter (Table 2)?

Probabilities in Table 2??? – should they be represented by a prob distribution w/ parameters?

Authors answer: We made the assumption, based on direct observations in the field that distribution of pork start always from the members of the household. This is another really interesting topic connecting wealth with exposure to TS infection risk. But unfortunately, again, we have not data to connect ability to pay with actual consume of pork. More studies are coming. The pork portion are given freely in the model without stratifying by household wealth. We explained this point better in the main text.

Reviewer response: OK (see above)

Minor (methods) comments:

Comment 3. The wording in the following sentence could be improved for clarity, and Significative effects should be rephrased to significant effect (line 154):

“Data from the 5 intervention villages, in which however, as showed in the original study (10), interventions produced no significative effects on observed HT and PC prevalence, were then used to validate the final calibration process”.

Authors answer: OK sentence rephrased to: “Data from the 5 intervention villages of the trial were used to validate the calibration process. As showed in the original study [10], the trial interventions produced no significative effects on observed HT and PC prevalence in those villages and the resulting empirical data can be considered as baseline unperturbed data exactly as the data from the 3 control villages. This makes the 5 intervention villages data the ideal candidate to validate the calibration process.”

Reviewer response: Thanks, the wording indicated above differs slightly from the revised wording in the manuscript (lines 170 – 176), please re-check this.

Results (pg.27)

Comment 11. While the authors highlight that the orders of magnitude are similar for the non-local parameters in the noNecro and simplified setups, which is true (Table 8), I am not convinced by the statement that the “simplified model calibration is able to estimate precisely the calibration parameters” (lines 574 – 575) – while they are not far off, particularly for the pigProglotIn probability there is quite a difference i.e. 3.86 to 11.23, so I think this statement is too strong to make. Are there uncertainty estimates available from the fitting procedure for each parameter, to then assess whether there is overlap in these uncertainty ranges (95% credible intervals) to make the comparison more valid? There is a reference missing on page 574, which might refer to Figure 7, in which case a stronger argument can be made for this statement?

Authors answer: Again, this problem of missing tables references. The reviewer is right this missing was a reference to Figure 7. Sorry again about that. The cited sentence “simplified model calibration is able to estimate precisely the calibration parameters” was did not referred to the actual value of the parameter but to the goodness of cross-validation test. We added an explanation to the main text to make this clearer.

Reviewer response: Thanks, can the authors include reference to “cross-correlation” in the Fig.7 legend since this is the terminology used on line 639

Comment #: About an uncertainty interval a comparison of parameter pigProglotInf for noNecro in figure 4 and for simplified in figure S3 fig 2 shows that the posterior distributions did not overlap at all so the two parameter estimation are truly different. We see this not as a problem because the model is different in the two setups. We agree that 3.86 is not strictly of the same order of magnitude of 11.23 so we changed the sentence in: “As expected, the calibrated model parametrization for the noNecro and simplified calibration setups (Table 8) are different with more pronounced differences in the values of parameters pHumanCyst and pigProglotInf”

Reviewer response: Thanks for this explanation, I think also highlighting the non-overlap with posterior distributions from the simplified calibration step (S3 fig 2) would be useful in the main text.

Comment 12. The authors state “The result of the simulations demonstrates that the parametrization obtained from the global calibration procedure presented here can be effectively exported to similar villages without any additional adjustment.” (lines 595 – 594) and while the average RE might similar, the variation in RE (particularly for higher REs) seems greater for the validated results to intervention villages (with max errors of 75% for HT (validated) compared to 45% for HT (calibrated) and 117% for PC (validated) compared to 75% for PC (calibrated)). I think the authors need to caveat the above statement, especially if they are interested in transferring the model to look at individual villages in a different setting. Is there for example a threshold difference in average RE that the authors would then the globally calibrated model would not be transferrable to another endemic setting?

Authors answer: We want to observe that the max error for HT validated is 37% and not 75%. That value falls close to the RE of 45% for the calibration. We agree with the reviewer about the fact that more quantitative criteria would be useful to discriminate among good and bad calibrations. Unfortunately, this would be possible only if we were able to estimate uncertainties of empirical data. With empirical data uncertainties it would be possible to say if a calibrated model parametrization is producing or not outcome in agreement with observed data. We are drafting a paper were pig seroprevalence is used as summary statistics and empirical data uncertainties on the observed data will make a mor quantitative criteria about model transferability available.

Reviewer response: Thanks for the further details. I am unclear why the max relative error is 37% (from village 510) rather than the RE from village 568 of 75% in table 9 to support the statement above. And is the average RE value of 45% for HT (for the control villages) calculated from Table 7? There is also a typo in line 656 (transferred to transferred)

Reviewer #2: The authors have made great improvement in this paper. Further, all comments and suggestions from all reviewers were addressed accordingly.

7. PLOS authors have the option to publish the peer review history of their article (what does this mean?). If published, this will include your full peer review and any attached files.

Reviewer #1: No

Reviewer #2: No

---

## [Author Response · Author response to Decision Letter 1]

8 Sep 2022

6. Review Comments to the Author

Reviewer #1: I provide follow-up comments below

Methods

Comment 1. What about seasonal differences in transmission (if influenced by the dry or 3-month rainy season indicated by the authors on lines 137-138). For example, Braae et al. 2014 found higher probability of free-ranging in the dry season compared to wet season and therefore different exposure risk (also possibly perturbing stable endemic transmission dynamics which most models currently assume). This study reflects the situation in Tanzania, a different endemic system, but may be relevant for this system.

Ref: Braae, U.C., Magnussen, P., Lekule, F. et al. Temporal fluctuations in the sero-prevalence of Taenia solium cysticercosis in pigs in Mbeya Region, Tanzania. Parasites Vectors 7, 574 (2014). https://doi.org/10.1186/s13071-014-0574-7

Authors answer: We agree. Seasonal differences are indeed important and, when possible, must be included in the model. Unfortunately, for the area under study, we have no data about seasonal changes of pig roaming areas (see ref 14). Other, possibly relevant changes in pork meat consumption can be connected for example with holydays and other social gatherings but also in this case no data are available. We deigned further studies to collect data on those topics that will be conducted in northern Peruvian rural areas.

Reviewer response: Thank you, it would be useful to include comment in the discussion on the potential role of seasonal dynamics, especially around changing pork consumption practices associated with holidays etc. Lightowlers & Donadeu 2017 also mention the importance of a “pork consumption calender”, so referencing this within the specific context of Northern Peru / development of the model would be really beneficial for the reader.

Authors answer 2: OK a comment was added to the discussion section of the manuscript about seasonal pattern of pork consumption in Peru you can find here below the sentence we added to the manuscript. 

Comment 2. The authors mention in lines 139 – 140 that “latrines…are often of poor quality and readily accessible by humans”. This seems like an important feature of this particular endemic system, so was information on latrine quality collected (alongside information on presence of latrine & adherence; line 158 & mapped on Fig 1) and incorporated into the model? Or do the authors assume that all latrines are of equally poor quality and therefore uniformly provide an exposure risk to free-roaming pigs?

If the latter (this seems the case, given lines 290-291 “The level of contamination of the defecation site depends on the presence of a latrine in the tapeworm carrier household and on adherence to its use”), this should be explicitly stated in the assumptions. 2

Authors answer: No information about latrines quality was collected during the field trial used to inform the model. We changed the text after lines 290 -291 to explain this better.

Reviewer response: Thank you for this additional information and text in the manuscript. So, you assumed all households had latrines in the “good state” based on the household census or you assumed this was the case?

Authors answer 2 : OKwe rephrased the sentence as: 

”During the household census, study teams characterized latrines generally as either being in good state or in disrepair with respect to the ability of pigs to have potential access to human feces. Within the model, we therefore assumed that when latrines characterized as being in a good state were used, these would completely prevent pig access to human feces.” 

Comment 4. The authors state new human agents are “periodically introduced” into the simulation (lines 192 – 193), with the rich dataset acquired through this study, would it not be feasible (and more realistic) to accurately simulate immigration introductions (for example are there differential rates during dry vs rainy seasons)?

Authors answer: Unfortunately, no data are available about human movements in the villages under study during the study period. For that reason, only rates from obtained from neighboring areas and from the entire Piura region were used un the study. No data about seasonal differences in human movements were available.

Reviewer response: Thank you, it would be useful for the authors to explicity state that human movement data was not available for the specific villages therefore migration rates were utilised from neighbouring areas (under the assumption that the rates are similar across villages in the Piura area?.

Authors answer 2: Ok new sentence added to the Methods – human module section: “Since human movement data was not available for the specific simulation villages, human movement rates were obtained from surveys conducted in neighboring areas under the assumption that the rates are similar across villages in the Piura area (see Table 3).”

Comment 5. “cysts being distributed randomly to the pork portions” (lines 201-202) is a strong assumption to include, there is good knowledge now on the carcass distribution of cysts (see Chembensofu et al. 2017), so could the authors indicate whether they have considered modelling this (assuming pork portions can be from different parts of the pig carcass?). The authors further state on lines 253 – 254 that “If the slaughtered pig is infected with T. solium, its cysts are distributed randomly to portions that are made of muscle, bones, and skin, but not to portions from entrails”; can the authors also explain here why cysts are not distributed to entrails (or a fuller description of what constitutes entrails would eb useful).

Ref: Chembensofu, M., Mwape, K.E., Van Damme, I. et al. Re-visiting the detection of porcine cysticercosis based on full carcass dissections of naturally Taenia solium infected pigs. Parasites Vectors 10, 572 (2017). https://doi.org/10.1186/s13071-017-2520-y

Authors answer: We totally agree: different part of the carcass present generally different cyst densities. However, this strong assumption is justified for two reasons: first in the villages under study carcass are divided into portions at home without following any predefined meat cut patterns. The second reason justifying our assumption is that even if the right distribution of cysts in different carcass parts were followed, we have no data to connecting consumption patterns and the individual characteristics of humans in the villages. The portions of pork containing different densities of cysts would be then distributed randomly to the villagers (following the geographical criteria indicated in the manuscript) resulting in no practical effect on transmission.

We based our assumption of no cysts in the entrails (it is now specified in the text that entrails are: liver, spleen, lungs and intestines) on papers like:

Boa ME, Kassuku AA, Willingham AL, Keyyu JD, Phiri IK, Nansen P. Distribution and density of cysticerci of Taenia solium by muscle groups and organs in naturally infected local finished pigs in Tanzania. Veterinary Parasitology. 2002;106: 155–164. doi:10.1016/S0304-4017(02)00037-7

That reported 0 cyst found in entrails. The paper cited in the comment reported 0 cysts in the entrails with the exception of the liver where only 3.7 % of cysts were found.

Reviewer response: Thank you, it would be useful for the authors to summarise the above justifications underpinning the assumption around lack of connecting portion allocations from different parts of the pig carcass to variable exposure riski.

Authors answer 2: OK we added to the manuscript the following new sentence: 

“It is known that different part of the carcass generally present different cyst densities [15]. However, the model assumption of uniform distribution of cysts in the muscle, bone and skin portions is justified by the fact that no data is available about the way different parts of the carcass are distributed in pork portions during at-home pig slaughtering. Additionally, there is no information about different rates of allocation of distinct parts of the carcass to people with dissimilar individual characteristics in the village. This lack of data makes not possible, in the model, to effectively represent the connection between variable risk of individual exposures and different densities of cysts in distinct pork portions obtained from the same slaughtered pig.”

Comment 8. After review of S1. Fig1. I am not clear where the decision process regarding sale or slaughter of pigs is included within the Household module flow chart. Indeed, it appears this is within the S2 Fig3: Pig module flow chart. Instead, which might cause some confusion when trying to match the description in the methods to these supplemental figures.

On a further note, S1 Fig 1 in the S2 word document should be S2 Fig 1? Can the authors check in detail throughout the supplementary to ensure there are no typos such as this please.

Authors answer: Thanks for noting that. We corrected accordingly figures S2 Fig1 and S2 Fig3 the pigs and household flow charts.

Reviewer response: Thank you for the revision; only minor suggestion would be to make sure the arrow on the line after the box “Designate breeding sow to be killed, replace sow” is before the intersection rather than after (otherwise appears that the decision tree does not continue to the pig selling boxes.

Author answer 2: OK figure corrected.

Comment 9. How valid is the assumption that “neighbouring areas have similar levels of T. solium transmission” regarding assigning the same probability of cysticercosis for imported pig agents (lines 240-242).

Authors answer: Unfortunately, this is only a guess because not studies on prevalence of pigs imported in the study rural villages were made. We then reasonably supposed that if imported from similar rural areas pigs had to present same levels of cysticercosis prevalence. More studies are coming.

Reviewer response: Thank you, I would suggest a short section in the discussion (or methods) to state where further studies will address these gaps/ assumptions (similar for the next comment on prevalence of imported pigs, number of distributed pork portions per household member) where the authors indicate more studies are coming.

Autors answer 2: OK. We added to the manuscript the following sentence:

Another process that is considered important in determining T. solium transmission and that was not represented in the model because of the limited amount of available data is the influence of seasonal dynamics on transmission. Furthermore, even if data are not available for the Piura district, for other areas in Peru an increase in pork consumption was observed at the end of the month of July, during the Peruvian national holidays and, to a limited extent, during Christmas holydays [14]. These changes, connected with cultural and religious practices, may have a strong impact on intervention effectiveness [27] and will be included in the model when more data will be available. 

 We anticipate conducting studies, in the Piura district, in the near future, on pig immunity, environmental contamination with eggs, human and pig, movements patterns relevant to T. solium transmission, infection prevalence of pigs imported into the villages, pattern of pork distribution to the village members.”

Comment 11. Furthermore, can the explain why pork portions are distributed more widely from the initial household (lines 256-260), are these pork portions sold to the other households, or given freely? If sold, is there a probability associated with the ability to pay for the recipient household, or is this effectively captured in the pigimportRateHousehold parameter (Table 2)?

Probabilities in Table 2??? – should they be represented by a prob distribution w/ parameters?

Authors answer: We made the assumption, based on direct observations in the field that distribution of pork start always from the members of the household. This is another really interesting topic connecting wealth with exposure to TS infection risk. But unfortunately, again, we have not data to connect ability to pay with actual consume of pork. More studies are coming. The pork portion are given freely in the model without stratifying by household wealth. We explained this point better in the main text.

Reviewer response: OK (see above)

Authors anwswer 2: OK (see above)

Minor (methods) comments:

Comment 3. The wording in the following sentence could be improved for clarity, and Significative effects should be rephrased to significant effect (line 154):

“Data from the 5 intervention villages, in which however, as showed in the original study (10), interventions produced no significative effects on observed HT and PC prevalence, were then used to validate the final calibration process”.

Authors answer: OK sentence rephrased to: “Data from the 5 intervention villages of the trial were used to validate the calibration process the model transfer process. We note that, Asas showed in the original study [10], the trial interventions produced no significative effects on observed HT and PC prevalence in those villages and therefore the resulting empirical data can be considered as measurements of the baseline unperturbed data villages state exactly as the data from the 3 control villages. This makes the 5 intervention villages data the ideal candidate to validate the calibration transfer process.”

Reviewer response: Thanks, the wording indicated above differs slightly from the revised wording in the manuscript (lines 170 – 176), please re-check this.

Authors answer 2: Sorry about that: the text is indeed a little different respect to what was reported here. Please find the actual manuscript sentence copied here below and the differences highlighted above.

Data from the 5 intervention villages of the trial were used to validate the model transfer process. We note that, as showed in the original study [10], the trial interventions produced no significative effects on observed HT and PC prevalence therefore the resulting empirical data can be considered as measurements of the baseline unperturbed villages state exactly as the data from the 3 control villages. This makes the 5 intervention villages data the ideal candidate to validate the transfer process. 

Results (pg.27)

Comment 11. While the authors highlight that the orders of magnitude are similar for the non-local parameters in the noNecro and simplified setups, which is true (Table 8), I am not convinced by the statement that the “simplified model calibration is able to estimate precisely the calibration parameters” (lines 574 – 575) – while they are not far off, particularly for the pigProglotIn probability there is quite a difference i.e. 3.86 to 11.23, so I think this statement is too strong to make. Are there uncertainty estimates available from the fitting procedure for each parameter, to then assess whether there is overlap in these uncertainty ranges (95% credible intervals) to make the comparison more valid? There is a reference missing on page 574, which might refer to Figure 7, in which case a stronger argument can be made for this statement?

Authors answer: Again, this problem of missing tables references. The reviewer is right this missing was a reference to Figure 7. Sorry again about that. The cited sentence “simplified model calibration is able to estimate precisely the calibration parameters” was did not referred to the actual value of the parameter but to the goodness of cross-validation test. We added an explanation to the main text to make this clearer.

Reviewer response: Thanks, can the authors include reference to “cross-correlation” in the Fig.7 legend since this is the terminology used on line 639

Authors answer 2: thanks for this comment: the term cross-correlation was eliminated from the manuscript to be substituted by: cross-validation

Comment #: About an uncertainty interval a comparison of parameter pigProglotInf for noNecro in figure 4 and for simplified in figure S3 fig 2 shows that the posterior distributions did not overlap at all so the two parameter estimation are truly different. We see this not as a problem because the model is different in the two setups. We agree that 3.86 is not strictly of the same order of magnitude of 11.23 so we changed the sentence in: “As expected, the calibrated model parametrization for the noNecro and simplified calibration setups (Table 8) are different with more pronounced differences in the values of parameters pHumanCyst and pigProglotInf”

Reviewer response: Thanks for this explanation, I think also highlighting the non-overlap with posterior distributions from the simplified calibration step (S3 fig 2) would be useful in the main text.

Authors answer 2: OK manuscript modified:

“The former parameter is associated with noNecro and simplified posterior distributions that do not overlap as it can be noted comparing Fig 4 and S3 Fig 2.”

Comment 12. The authors state “The result of the simulations demonstrates that the parametrization obtained from the global calibration procedure presented here can be effectively exported to similar villages without any additional adjustment.” (lines 595 – 594) and while the average RE might similar, the variation in RE (particularly for higher REs) seems greater for the validated results to intervention villages (with max errors of 75% for HT (validated) compared to 45% for HT (calibrated) and 117% for PC (validated) compared to 75% for PC (calibrated)). I think the authors need to caveat the above statement, especially if they are interested in transferring the model to look at individual villages in a different setting. Is there for example a threshold difference in average RE that the authors would then the globally calibrated model would not be transferrable to another endemic setting?

Authors answer: We want to observe that the max error for HT validated is 37% and not 75%. That value falls close to the RE of 45% for the calibration. We agree with the reviewer about the fact that more quantitative criteria would be useful to discriminate among good and bad calibrations. Unfortunately, this would be possible only if we were able to estimate uncertainties of empirical data. With empirical data uncertainties it would be possible to say if a calibrated model parametrization is producing or not outcome in agreement with observed data. We are drafting a paper were pig seroprevalence is used as summary statistics and empirical data uncertainties on the observed data will make a mor quantitative criteria about model transferability available.

Reviewer response: Thanks for the further details. I am unclear why the max relative error is 37% (from village 510) rather than the RE from village 568 of 75% in table 9 to support the statement above. And is the average RE value of 45% for HT (for the control villages) calculated from Table 7? There is also a typo in line 656 (transferred to transferred)

Authors answer 2: Thanks for the comment: it is right the max relative error is 75%. However, there are no wrong numbers were presented in the manuscript. The average RE for HT of control villages mentioned in the manuscript is actually 40% and not 45%: “). These results are very close to the average REs of control villages (40% and 45% for HT and PC, respectively). :” and yes, it is calculated from Table 7. This is now specified in the manuscript text: “, see Table 7”

Reviewer #2: The authors have made great improvement in this paper. Further, all comments and suggestions from all reviewers were addressed accordingly.

7. PLOS authors have the option to publish the peer review history of their article (what does this mean?). If published, this will include your full peer review and any attached files.

Do you want your identity to be public for this peer review? For information about this choice, including consent withdrawal, please see our Privacy Policy.

Reviewer #1: No

Reviewer #2: No

---

## [Editor Report · Decision Letter 2]

13 Sep 2022

Non-local validated parametrization of an agent-based model of local-scale Taenia solium transmission  in North-West Peru

PONE-D-21-27051R2

Dear Dr. Pizzitutti,

We’re pleased to inform you that your manuscript has been judged scientifically suitable for publication and will be formally accepted for publication once it meets all outstanding technical requirements.

Kind regards,

Marcello Otake Sato, Ph.D., D.V.M.

Academic Editor

PLOS ONE
---

## [Editor Report · Acceptance letter]

19 Sep 2022

PONE-D-21-27051R2 

Non-local validated parametrization of an agent-based model of local-scale *Taenia solium* transmission   in North-West Peru 

Dear Dr. Pizzitutti:

I'm pleased to inform you that your manuscript has been deemed suitable for publication in PLOS ONE. Congratulations! Your manuscript is now with our production department. 

Kind regards, 

on behalf of

Dr. Marcello Otake Sato 

Academic Editor

PLOS ONE